# ADAPTIVE MARGIN RLHF VIA PREFERENCE OVER PREFERENCES

## ABSTRACT

Margin-based optimization is fundamental to improving generalization and robustness in classification tasks. In the context of reward model learning from preferences within Reinforcement Learning from Human Feedback (RLHF), existing methods typically rely on no margins, fixed margins, or margins that are simplistic functions of preference ratings. However, such formulations often fail to account for the varying strengths of different preferences—i.e., some preferences are associated with larger margins between responses—or they rely on noisy margin information derived from preference ratings. In this work, we argue that modeling the strength of preferences can lead to better generalization and more faithful alignment. Furthermore, many existing methods that use adaptive margins assume access to accurate preference scores, which can be difficult for humans to provide reliably. We propose a novel approach that leverages preferences over preferences—that is, annotations indicating which of two preferences reflects a stronger distinction. We use this ordinal signal to infer adaptive margins on a per-datapoint basis. We introduce an extension to Direct Preference Optimization (DPO), DPO-PoP, that incorporates adaptive margins from preference-over-preference supervision, enabling improved discriminative and generative performance. Empirically, our method improves over vanilla DPO, DPO with fixed margins, and DPO with ground-truth margins on the UltraFeedback dataset. These results suggest that integrating preference-over-preference information, which requires less precision to be provided accurately, can improve discriminative and generative performance without adding significant complexity. Additionally, we show that there is a tradeoff between discriminative and generative performance: improving test classification accuracy, particularly by correctly labeling weaker preferences at the expense of stronger ones, can lead to a decline in generative quality. To navigate this tradeoff, we propose two sampling strategies to gather preference-over-preference labels: one favoring discriminative performance and one favoring generative performance.

## 1 INTRODUCTION

Margin-based approaches have been pivotal in the design and analysis of classification algorithms. In classical machine learning, the margin, defined as the distance between a decision boundary and data points, acts as a proxy for confidence and plays a critical role in improving generalization. For example, Support Vector Machines (SVMs) explicitly maximize the minimum margin, which has been shown to enhance robustness and reduce overfitting (Cortes & Vapnik, 1995). Ensemble methods like AdaBoost (Freund et al., 1996) also leverage margin-based generalization, as boosting algorithms implicitly seek to increase the margin distribution across training samples (Schapire et al., 1998).

Although fixed-margin strategies have proven effective, they assume fixed and equal margin for all training data points. This has motivated the development of adaptive margin approaches, where the margin varies across examples based on criteria such as sample difficulty, uncertainty, or class imbalance. Adaptive Margin SVMs (Herbrich & Weston, 1999) use different margin values for different training data points and provide bounds on the generalization error, justifying its robustness against outliers. Furthermore, methods such as CurricularFace (Huang et al., 2020), AdaCos (Zhang et al., 2019), and adaptive triplet losses (Ha & Blanz, 2021) have shown that adapting the margin dynamically during training leads to more stable optimization and better generalization, particularly in settings such as face recognition or imbalanced classification.

In Reinforcement Learning from Human Feedback (RLHF), pairwise preference data from humans is used to learn a reward function or policy. The Bradley-Terry (BT) model (Bradley & Terry, 1952) is widely used to model pairwise preference data, where the probability of preferring one output over another is determined by the difference in their reward scores. This preference model is commonly used in the alignment of large language models (LLMs) (Ouyang et al., 2022; Touvron et al., 2023), in which a reward function is learned to rank outputs based on human preferences, and subsequently used to optimize the policy.

Current reward modeling approaches generally fall into two categories. Some methods treat all preferences equally by applying no margin at all (Ouyang et al., 2022). Others incorporate unequal treatment by introducing adaptive margins, which are typically derived in one of two ways: either from scalar scores assigned to preferences by human annotators or language models (Touvron et al., 2023; Wang et al., 2025), or from the outputs of learned reward models (Wang et al., 2024a; Qin et al., 2024; Amini et al., 2024; Wang et al., 2024b). Using constant or no margin information fails to account for the varying strength of different preferences—that is, the degree to which one response is favored over another within a given preference. Obtaining preference strength information from preference scores, allows us to use adaptive margin information, but requires us to collect scalar feedback from LLMs or humans.

Specifying preference strength typically requires a numerical score, which may be difficult for humans to provide accurately. For instance, when using labeling schemes such as Likert ratings, where annotators rate responses individually rather than comparatively, the scores may not be consistently calibrated. That is, even if annotators agree on which response is better in a pair, they may assign inconsistent scores due to differences in how they interpret the scale (Wadhwa et al., 2024). By contrast, preference over preference annotation requires less precision to be provided accurately, compared to assigning scores to individual responses. Comparitive annotation, particularly Best-to-Worst scaling (BWS), has been to shown to produce significantly more reliable results than rating scale annotations such as Likert scales (Kiritchenko & Mohammad, 2017; Burton et al., 2019). BWS also demonstrated greater reliability when applied to linguistically complex cases, such as phrases containing negation or modals (Kiritchenko & Mohammad, 2017). Best-to-Worst scaling (BWS) is an extension of Thurstone's method of paired comparisons (Thurstone, 2017) which is another paired comparison statistical model like Bradley-Terry (Bradley & Terry, 1952; Handley, 2001) We use this as a motivation to propose preference over preference (PoP) labeling, in which annotators compare two preferences and indicate which one reflects a stronger preference. Rather than assigning scores to individual responses (Cui et al., 2024; Wang et al., 2023), in our preference-over-preference setting, annotators compare preference pairs and select the pair for which the contrast between the chosen and rejected responses is more pronounced. More importantly, preference-over-preferences allow us to infer continuous real-valued margins for preferences, compared to rating scale annotations, which only offer discrete numerical options. Using this PoP supervision, we construct a dataset of preference over preference comparisons that enables us to infer adaptive margin information for each datapoint.

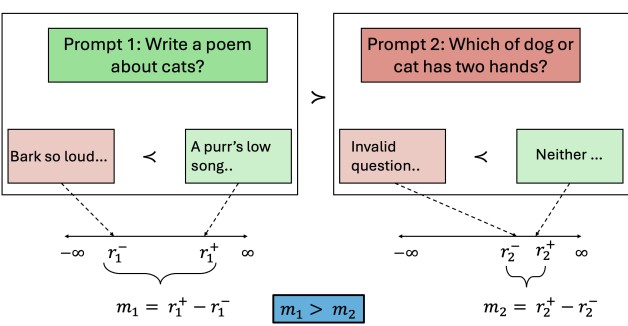

Figure 1: A pictorial illustration of the PoP framework. A preference is stronger than another when the reward difference between its preferred and dispreferred responses is larger. The reward difference of the weaker preference in the pair serves as the margin for the stronger preference.

In this work, we propose DPO-PoP, an alignment algorithm that integrates preference-over-preference (PoP) supervision into the Direct Preference Optimization (DPO) framework (Rafailov et al., 2024b), enabling margin-aware alignment of large language models (LLMs) with human preferences using only supervised learning. For each data point, we use PoP supervision to infer an adaptive margin that reflects the relative strength of the underlying preference. A pictorial illustration of the PoP framework is presented in Figure 1. We demonstrate that collecting PoP supervision is a simple and effective way to improve both the discriminative and generative performance of LLMs. Our

results show that DPO-PoP variants improve over all baselines in both respects. Moreover, we highlight a tradeoff between discriminative performance, as measured by test classification accuracy, and generative performance, as measured by win rate—where improving classification accuracy on weaker preferences at the expense of stronger ones—can lead to a decline in generative quality. To navigate this tradeoff, we propose two sampling strategies for generating preference-over-preference labels: iterative sampling, which favors discriminative performance, and random sampling, which favors generative performance.

## 2 BACKGROUND

### 2.1 REWARD MODELING

In the reward modeling stage of Reinforcement Learning from Human Feedback (RLHF), a reward model is trained to assign scalar scores to prompt-response pairs, indicating how well a response aligns with human preferences. This process relies on a preference dataset $\mathcal{D}_{\text{pref}} = (x_i, y_i^+, y_i^-)_{i=1}^N$, where $x_i$ is a prompt, $y_i^+$ is the preferred response, and $y_i^-$ is the dispreferred response. The Bradley-Terry (BT) model (Bradley & Terry, 1952) is commonly used to model preference likelihoods.

$$P(y^+ \succ y^-) = \frac{e^{r(x,y^+)}}{e^{r(x,y^+)} + e^{r(x,y^-)}} = \sigma(r(x,y^+) - r(x,y^-)) \tag{1}$$

Here, $r$ denotes the reward assigned to a prompt-response pair, and $\sigma$ denotes the sigmoid function. We parameterize the reward function as $r_\phi$, and use it to approximate the ground-truth reward function by maximizing the likelihood of the observed preference data under the Bradley-Terry model. For more details on the RLHF pipeline, refer to Appendix $C$

$$\min_\phi -\mathbb{E}_{(x,y^+,y^-)\sim\mathcal{D}_{\text{pref}}}[\log \sigma(r_\phi(x,y^+) - r_\phi(x,y^-))] \tag{2}$$

### 2.2 DIRECT PREFERENCE OPTIMIZATION

Direct Preference Optimization (DPO) (Rafailov et al., 2024b) belongs to a class of algorithms, called Direct Alignment Algorithms (DAAs) (Rafailov et al., 2024a), which aim to directly align a policy from preference data via supervised learning, without having to learn a reward model or use reinforcement learning. DPO utilizes the closed form solution of the optimal KL regularized reward policy (Peters & Schaal, 2007; Peng et al., 2019), and expresses the rewards in the Bradley-Terry preference model (Bradley & Terry, 1952), directly in terms of the optimal policy. This allows us to learn a parameterized optimal policy directly from the preference data, using Equation 3

$$\mathcal{L}_{DPO}(\pi_\theta; \pi_{ref}) = \mathbb{E}_{(x,y^+,y^-)\sim\mathcal{D}_{\text{pref}}}\left[-\log \sigma\left(\beta \log \frac{\pi_\theta(y^+|x)}{\pi_{\text{ref}}(y^+|x)} - \beta \log \frac{\pi_\theta(y^-|x)}{\pi_{\text{ref}}(y^-|x)}\right)\right] \tag{3}$$

The implicit reward assigned by the DPO model to a response $y$ given a prompt $x$ is $\beta \log \frac{\pi_\theta(y|x)}{\pi_{\text{ref}}(y|x)}$.

### 2.3 MARGINS IN REWARD MODELING

Margins can be incorporated into the reward modeling phase of the RLHF pipeline to enforce not only that the reward model ranks the preferred response higher than the dispreferred one, but also that it assigns a sufficiently large difference in reward scores—either through fixed or adaptive margins. The margin-based reward modeling loss can be expressed as:

$$\min_\phi -\mathbb{E}_{(x,y^+,y^-)\sim\mathcal{D}_{\text{pref}}}[\log \sigma(r_\phi(x,y^+) - r_\phi(x,y^-) - m(x,y^+,y^-)] \tag{4}$$

Here $m(x,y^+,y^-)$ denotes the margin term. In the fixed margin setting this can be a constant. In the adaptive-margin setting, it can be defined as a function of the preference instance, for example, based on the degree of discrepancy between the preferred and dispreferred responses.

## 3   METHOD: ADAPTIVE MARGIN DPO WITH PREFERENCES OVER PREFERENCES

To obtain adaptive margin information, in which each preference datapoint is assigned a different margin, and stronger preferences are associated with larger margins than weaker ones, we propose preferences over preferences (PoP) supervision. Given two standard preference comparisons, such as $A \succ B$ and $C \succ D$, we collect a label indicating which of the two preferences is stronger, from a labeler. For example, if the supervision indicates that $(A \succ B) \succ (C \succ D)$, this means that the discrepancy between $A$ and $B$ is greater than that between $C$ and $D$ under the ground-truth reward function $r$. Formally, this implies:

$$r(A) - r(B) > r(C) - r(D)$$

This insight allows us to treat the margin from the weaker preference (e.g., $r(C) - r(D)$) as a lower bound on the margin for the stronger preference (e.g., $A \succ B$). Rather than regressing to a specific value, we enforce that the margin for the stronger preference must be at least as large as that of the weaker one.

We assume access to a dataset of preference over preference examples:

$$\mathcal{D}_{\text{PoP}} = \left\{ \left( (x_{s_i}, y_{s_i}^+, y_{s_i}^-), \ (x_{w_i}, y_{w_i}^+, y_{w_i}^-) \right) \right\}_{i=1}^N$$

Here, $(x_{s_i}, y_{s_i}^+, y_{s_i}^-)$ represents the stronger preference in the pair, where $x_{s_i}$ is the prompt, $y_{s_i}^+$ is the preferred response, and $y_{s_i}^-$ is the dispreferred response. Similarly, $(x_{w_i}, y_{w_i}^+, y_{w_i}^-)$ denotes the weaker preference, where $x_{w_i}$ is the prompt, $y_{w_i}^+$ is the preferred response, and $y_{w_i}^-$ is the dispreferred response. Note that, unlike in standard reward modeling datasets, the prompts $x_{s_i}$ and $x_{w_i}$ can differ within a single PoP example, as PoP supervision compares the strength of entire preference instances, not individual responses.

We can express the adaptive margin reward modelling objective on a dataset of preferences over preferences as follows

$$\min_\phi \ \mathbb{E}_{\mathcal{D}_{\text{PoP}}} \Big[ - \log \sigma \big( r_\phi(x_s, y_s^+) - r_\phi(x_s, y_s^-) $$
$$- \text{sg} \left[ r_\phi(x_w, y_w^+) - r_\phi(x_w, y_w^-) \right] \big) \Big] \qquad (5)$$

Here, $\text{sg}[\cdot]$ denotes the stop-gradient operator. Although the adaptive margin is computed using the reward model $r_\phi$, we treat the margin derived from the weaker preference as a *fixed reference* during optimization. Applying the stop-gradient operator ensures that gradients do not propagate through this margin term, thereby preventing it from influencing updates to the reward model parameters $\phi$. Without the stop-gradient operator, the objective would incentivize parameters that invert the weaker preference to minimize the loss.

We use the closed-form solution for the optimal policy of a KL regularized reward problem to express the rewards directly in terms of the optimal policy, as in DPO (Rafailov et al., 2024b). Parameterizing the optimal policy by $\theta$, we end up with the DPO Preference-over-Preference loss

$$\min_\theta \ \mathbb{E}_{\mathcal{D}_{\text{PoP}}} \left[ - \log \sigma \left( \beta \left( \log \frac{\pi_\theta(y_s^+ \mid x_s)}{\pi_{\text{ref}}(y_s^+ \mid x_s)} - \log \frac{\pi_\theta(y_s^- \mid x_s)}{\pi_{\text{ref}}(y_s^- \mid x_s)} \right) \right. \right.$$
$$\left. \left. - \text{sg} \left[ \beta \left( \log \frac{\pi_\theta(y_w^+ \mid x_w)}{\pi_{\text{ref}}(y_w^+ \mid x_w)} - \log \frac{\pi_\theta(y_w^- \mid x_w)}{\pi_{\text{ref}}(y_w^- \mid x_w)} \right) \right] \right) \right] \qquad (6)$$

The DPO Preference-over-Preference (DPO-PoP) objective enables margin-aware alignment directly from PoP data using supervised learning, without requiring an explicit reward modeling stage or reinforcement learning. However, Equation 6 suffers from unstable gradients due to unbounded margins, resulting in a rapidly fluctuating loss that can explode during training. To mitigate this, we clip the margin values to lie within a fixed interval $[0, M_{\max}]$, where $M_{\max}$ is a user-specified constant. Margin values outside this range are clipped to the nearest endpoint, using a clipping

function $\text{clip}_{[0, M_{\max}]}$, which improves optimization stability. Additionally, to further stabilize training, we compute the margins using a slowly-updated target policy $\pi_{\hat{\theta}}$, whose parameters $\hat{\theta}$ track the policy $\pi$ via Polyak averaging over the model parameters $\theta$. This prevents the margin estimates from changing too rapidly across training steps. With these modifications, our final DPO-PoP objective is given by Equation 7

$$\min_{\theta} \ \mathbb{E}_{\mathcal{D}_{\text{PoP}}} \left[ -\log \sigma \left( \beta \left( \log \frac{\pi_{\theta}(y_s^+ \mid x_s)}{\pi_{\text{ref}}(y_s^+ \mid x_s)} - \log \frac{\pi_{\theta}(y_s^- \mid x_s)}{\pi_{\text{ref}}(y_s^- \mid x_s)} \right) \right. \right.$$

$$\left. \left. - \text{sg} \left[ \text{clip}_{[0, M_{max}]} \left( \beta \left( \log \frac{\pi_{\hat{\theta}}(y_w^+ \mid x_w)}{\pi_{\text{ref}}(y_w^+ \mid x_w)} - \log \frac{\pi_{\hat{\theta}}(y_w^- \mid x_w)}{\pi_{\text{ref}}(y_w^- \mid x_w)} \right) \right) \right] \right) \right] \quad (7)$$

## 4 RESULTS

We focus on the following research questions: **[Q1]** Does using DPO-PoP lead to models with improved discriminative ability? **[Q2]** Does using DPO-PoP lead to models with improved generative ability? We investigate these questions by evaluating the performance of our models on the test split of the UltraFeedback dataset (Cui et al., 2024) and external benchmarks such as RewardBench (Lambert et al., 2024) and AlpacaEval-2 (Dubois et al., 2025). More importantly, we also investigate **[Q3]**: Do the same trends observed in Q1 and Q2 hold when PoP annotations are gathered from an LLM annotator? This is important because it sheds light on whether PoP annotation is a practically viable alternative to rating-scale annotations for improving performance.

### 4.1 SYNTHETIC DATA EXPERIMENTS

#### 4.1.1 GENERATING THE PREFERENCE OVER PREFERENCE DATA

We use the UltraFeedback (Cui et al., 2024) binarized dataset[1] for our evaluations. The dataset provides scalar scores for the chosen and rejected responses, aggregated from multiple LLM evaluators. We compute the ground-truth margin for each preference as the score difference between the two responses, which also enables construction of PoP comparisons. Although a preference dataset of size $|D_{\text{pref}}|$ can yield up to $\frac{|D_{\text{pref}}|(|D_{\text{pref}}|-1)}{2}$ PoP pairs, we restrict the PoP dataset to $|D_{\text{PoP}}| = k|D_{\text{pref}}|$ to keep it manageable. Appendix E provides justification for using smaller values of $k$ and analyzes performance as a function of $k$; we use $k = 2$ by default. We also exclude pairs whose margin differences are below one, as they represent nearly indistinguishable preferences.

We evaluate two strategies for constructing the PoP dataset: one that represents each preference from the original dataset equally, and one that represents preferences in proportion to preference strength. We do this to explore the impact of different sampling strategies used to generate the PoP dataset, on downstream discriminative and generative performance. In the **iterative sampling** approach, each preference data point is equally represented by comparing it against $k$ weaker preferences (as judged by their margins). In practice, without ground-truth margin data, we could choose a preference and provide comparison preferences, asking the user for a label. We only choose $k$ preference pairs in which our chosen preference is judged to be stronger than the comparative preference. In contrast, the **random sampling** approach constructs the PoP dataset by randomly selecting pairs of preferences and labeling them based on their margins. This results in stronger preferences appearing more frequently in the PoP dataset than weaker ones. Furthermore, the **random sampling** approach is straightforward to implement in practice, in comparison to the iterative sampling approach, as this would only involve randomly sampling pairs of preferences and asking the annotator for a label. After generating the PoP dataset, we discard the original scalar scores and do not use them at any stage of model training.

#### 4.1.2 EXPERIMENTAL SETUP

We consider two models in our experiments: Llama-3.2-3b and Llama-3.1-8b (Grattafiori et al., 2024). Following the standard direct alignment pipeline, we align these models using the UltraFeedback preference dataset (Cui et al., 2024). We begin with a pretrained model and fine-tune it on the

---

[1] `HuggingFaceH4/ultrafeedback_binarized`

supervised fine-tuning (SFT) partition of the UltraFeedback dataset. Next, we align the models using the preference data from the same dataset. For further experimental details, refer to Appendix B

We evaluate the following variants of Direct Preference Optimization (DPO):

1. **Vanilla DPO**: No margin is used in the loss function.

2. **DPO-margin-1**: A fixed margin of 1 is applied to all preferences.

3. **DPO-margin-gt**: Ground-truth margin values from the UltraFeedback dataset are used.

4. **DPO-margin-gt-scaled**: This corresponds to the Scaled Bradley-Terry loss from Wang et al. (2025). The loss incorporates ground-truth margin information outside the log-sigmoid function rather than inside, effectively placing greater weight on preferences with larger margins. This can be interpreted as repeatedly sampling stronger preferences. The loss is defined as:
$$\mathcal{L}_{\text{SBT}} = -m \log \sigma \left( \beta \log \frac{\pi_\theta(y^+|x)}{\pi_{\text{ref}}(y^+|x)} - \beta \log \frac{\pi_\theta(y^-|x)}{\pi_{\text{ref}}(y^-|x)} \right) \tag{8}$$

5. **DPO-PoP-iter**: Margins are inferred from preference-over-preference (PoP) supervision, using a PoP dataset constructed via iterative sampling.

6. **DPO-PoP-random**: Margins are inferred from PoP supervision, using a PoP dataset constructed via random sampling. This strategy can be interpreted as a bootstrapped version of the loss employed in DPO-margin-gt-scaled, along with a margin term (inside the log-sigmoid) that is inferred from preference-over-preference supervision.

We provide the results for Llama-3.2-3b here. Results for Llama-3.1-8b are provided in Appendix D

### 4.1.3 DISCRIMINATIVE ABILITY

We evaluate DPO-PoP's discriminative ability and margin correlation. For each preference $A \succ B$, we compare the UltraFeedback score difference (ground truth) with the DPO implicit reward difference (prediction). High correlation indicates better generalization and calibrated preference strength estimation. We report both Spearman and Pearson correlations. The correlation metrics are only possible in this setting due to access to UltraFeedback scores and cannot be computed when PoP labels are annotator-generated; this analysis is provided purely for insight.

Table 1 shows that DPO-PoP-Iter attains the best test classification accuracy, outperforming even DPO-margin-gt, despite the latter having access to the true margin values.

The correlation metrics tell a different story: DPO-PoP-Random achieves the strongest Spearman and Pearson correlations, with DPO-PoP-Iter performing similarly on Spearman but substantially worse on Pearson. This suggests that DPO-PoP-Iter captures the correct ranking of preferences but its predicted margins are nonlinearly related to the true ones.

We also see that DPO-PoP-Random exhibits lower accuracy but higher correlations overall. Figure 2 explains this tradeoff: DPO-PoP-Iter correctly classifies more weak-preference examples, boosting accuracy, whereas DPO-PoP-Random better captures strong preferences and is less influenced by noisy weak comparisons. As a result, DPO-PoP-Random maintains more faithful linear and ordinal relationships to the ground-truth margins, yielding superior Pearson and Spearman correlations.

We also report performance on RewardBench (Lambert et al., 2024) in Table 2. The DPO-PoP variants outperform all baselines, including those with access to ground-truth margins. Examining the Overall score, we observe that DPO-PoP-random achieves the highest performance. Notably, DPO-PoP-iter heavily outperforms all methods on the Chat split but also strongly underperforms on the Reasoning split—which comprises a larger portion of the dataset—resulting in a lower Overall score compared to DPO-PoP-random. In contrast, DPO-PoP-random delivers stable performance across all categories, securing the highest Overall score.

### 4.1.4 GENERATIVE ABILITY

Next, we use UltraRM (Cui et al., 2024) to evaluate the responses of each of the aligned models and compare the quality of their generations. We use Vanilla-DPO as the reference model against which the other DPO variants are judged. We calculate the win rate and the median advantage of each model vs Vanilla DPO, as judged by UltraRM. The advantage of a datapoint is the difference between the UltraRM rewards of the response generated by the test model and the reference model, for a given prompt. The median advantage of a model is computed as the median of these per-prompt advantages

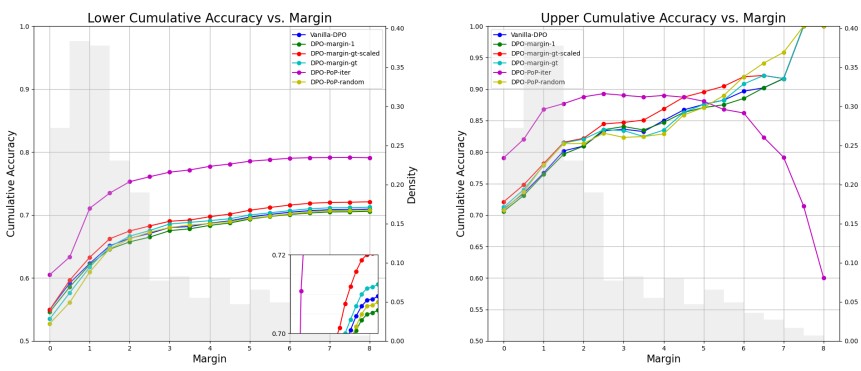

((a)) Lower Cumulative Accuracy vs Margin     ((b)) Upper Cumulative Accuracy vs Margin

Figure 2: Cumulative Accuracy vs Margin for the different DPO variants considered. Lower Cumulative Accuracy at margin $m$ indicates the accuracy of predicting preference labels using only datapoints with ground-truth margin less than or equal to $m$. Conversely, Upper Cumulative Accuracy reflects prediction accuracy on datapoints with ground-truth margin greater than or equal to $m$. The dark grey histogram shows the distribution (density) of margin values in the test set. In plot (a), DPO-PoP-Iter achieves higher accuracy on datapoints with lower margins, while in plot (b), its performance drops for higher margin datapoints. The lower cumulative accuracy plot is zoomed in, to address a reviewers request.

| Algorithm | Pearson Correlation | Spearman Correlation | Accuracy (%) |
|---|---|---|---|
| Vanilla-DPO | $0.2940 \pm 0.0036$ | $0.3003 \pm 0.0036$ | $71.15 \pm 0.178$ |
| DPO-margin-1 | $0.2929 \pm 0.0041$ | $0.2984 \pm 0.0045$ | $7118 \pm 0.28$ |
| DPO-margin-gt | $0.3427 \pm 0.0029$ | $0.3451 \pm 0.0028$ | $71.85 \pm 0.34$ |
| DPO-margin-gt-scaled | $0.3381 \pm 0.0037$ | $0.3453 \pm 0.0033$ | $72.05 \pm 0.16$ |
| DPO-PoP-iter | $0.2449 \pm 0.0017$ | $\mathbf{0.3656 \pm 0.0008}$ | $\mathbf{79.97 \pm 0.41}$ |
| DPO-PoP-random | $\mathbf{0.3639 \pm 0.0020}$ | $\underline{0.3685 \pm 0.0010}$ | $\underline{71.09 \pm 0.21}$ |

Table 1: Comparison of DPO variants on classification accuracy and Spearman, Pearson correlation with ground-truth margins for Llama-3.2-3b. This table was modified to include confidence intervals over 6 seeds (including the earlier result) to address the reviewers' questions during the rebuttals.

over the entire test set. The results are displayed in the Table 3. We observe that DPO-PoP-random outperforms all other baselines in terms of win rate and median advantage. DPO-PoP-random which infers margins from PoP supervision, outperforms DPO variants that have access to ground truth margins.

We also report the performance of all the DPO variants on the AlpacaEval 2.0 benchmark (Dubois et al., 2025) in Table 4. DPO-PoP-random outperforms all other baselines both in terms of win-rate and length controlled win-rate.

In both Tables 3 and 4, we observe that DPO-PoP-iter underperforms compared to DPO-PoP-random and DPO-margin-gt. We hypothesize that this is due to correctly classifying weaker preferences at the expense of stronger preferences, as discussed in Section 4.1.3. By potentially overfitting to noisy weaker preferences, DPO-PoP-iter suffers a drop in generative performance.

### 4.2 LLM ANNOTATED PREFERENCE OVER PREFERENCE DATA EXPERIMENTS

Instead of using the margin information from the UltraFeedback dataset (Cui et al., 2024) to infer Preference-over-Preference (PoP) labels, we directly obtain PoP annotations from an LLM (GPT-4.1-mini). This setup serves as a test bed for evaluating PoP-based methods in realistic settings, where PoP labels would typically come from either LLM or human annotators.

To keep annotation cost low, we begin by randomly sampling 5,000 preference examples from UltraFeedback. This subset is used to train all baseline models. To construct the PoP dataset, we then

| Algorithm | Chat | Chat Hard | Safety | Reasoning | Overall |
|---|---|---|---|---|---|
| Vanilla-DPO | $75.65 \pm 0.34$ | $\mathbf{64.51} \pm \mathbf{0.51}$ | $71.49 \pm 0.17$ | $75.85 \pm 0.46$ | $75.46 \pm 0.21$ |
| DPO-margin-1 | $76.86 \pm 0.54$ | $\underline{64.14} \pm 0.21$ | $71.19 \pm 0.86$ | $77.03 \pm 0.23$ | $75.78 \pm 0.29$ |
| DPO-margin-gt | $80.35 \pm 0.38$ | $63.27 \pm 0.21$ | $75.70 \pm 0.31$ | $\mathbf{78.05} \pm \mathbf{0.47}$ | $77.45 \pm 0.25$ |
| DPO-margin-gt-scaled | $80.87 \pm 0.55$ | $64.11 \pm 0.53$ | $75.47 \pm 0.46$ | $\underline{76.33} \pm 0.27$ | $77.13 \pm 0.29$ |
| DPO-PoP-iter | $\mathbf{87.71} \pm \mathbf{0.53}$ | $59.61 \pm 0.50$ | $81.28 \pm 0.62$ | $69.83 \pm 1.35$ | $76.73 \pm 0.24$ |
| DPO-PoP-random | $\underline{82.73} \pm 0.80$ | $62.54 \pm 0.63$ | $\mathbf{81.94} \pm \mathbf{1.07}$ | $76.44 \pm 0.69$ | $\mathbf{78.87} \pm \mathbf{0.25}$ |

Table 2: Performance of Llama-3.2-3b DPO variants on RewardBench. Higher is better. This table was modified to include confidence intervals over 6 seeds (including the earlier result) to address the reviewers' questions during the rebuttals.

| Method | Median Advantage | Win Rate (%) |
|---|---|---|
| DPO-margin-1 | $0.2272 \pm 0.0202$ | $54.91 \pm 0.34$ |
| DPO-margin-gt | $0.5863 \pm 0.0577$ | $61.25 \pm 1.15$ |
| DPO-margin-gt-scaled | $0.1602 \pm 0.0284$ | $53.65 \pm 0.64$ |
| DPO-PoP-iter | $0.3887 \pm 0.0452$ | $57.76 \pm 0.88$ |
| DPO-PoP-random | $\mathbf{0.6745} \pm \mathbf{0.0506}$ | $\mathbf{62.39} \pm \mathbf{1.12}$ |

Table 3: Comparison of margin-based DPO variants against Vanilla DPO on median advantage and win rate for Llama-3.2-3b. This table was modified to include confidence intervals over 6 seeds (including the earlier result) to address the reviewers' questions during the rebuttals.

| Experiment | Length-Controlled Win Rate | Win Rate | Avg Length |
|---|---|---|---|
| Vanilla-DPO | $11.74 \pm 0.74$ | $11.37 \pm 0.69$ | $1800 \pm 17$ |
| DPO-margin-1 | $11.74 \pm 1.04$ | $11.51 \pm 1.04$ | $1823 \pm 29$ |
| DPO-margin-gt | $12.40 \pm 0.71$ | $12.17 \pm 0.58$ | $1915 \pm 42$ |
| DPO-margin-gt-scaled | $10.99 \pm 0.79$ | $10.97 \pm 0.71$ | $1836 \pm 19$ |
| DPO-PoP-iter | $12.30 \pm 0.70$ | $12.26 \pm 0.62$ | $1919 \pm 50$ |
| DPO-PoP-random | $\mathbf{14.24} \pm \mathbf{1.06}$ | $\mathbf{13.69} \pm \mathbf{1.02}$ | $1846 \pm 20$ |

Table 4: Performance of Llama-3.2-3b DPO variants on the AlpacaEval 2.0 benchmark. This table was modified to include confidence intervals over 6 seeds (including the earlier result) to address the reviewers' questions during the rebuttals.

sample random pairs of preferences from this subset and ask the LLM to identify which preference in each pair is stronger. The resulting LLM-annotated PoP dataset is used to train DPO-PoP-Random. We focus on the Random variant because PoP annotations are far easier to obtain in this setting than those required for DPO-PoP-Iter. Following the setup in the synthetic data experiments, we use $k = 2$ and use the Llama3.2-3b model for our experiments. Additional experiments showing how performance of DPO-PoP algorithms is impacted by preference-over-preference labeling noise are provided in Appendix F. We also provide the prompt used to gather POP annotations from an LLM in Appendix K.

### 4.2.1 DISCRIMINATIVE PERFORMANCE

The results showing the test classification accuracy on the UltraFeedback dataset (Cui et al., 2024) and RewardBench (Lambert et al., 2024) scores are in Tables 5 and 6 respectively.

### 4.2.2 GENERATIVE PERFORMANCE

The results displaying the win rate of the model responses as judged by UltraRM (Cui et al., 2024) and AlpacaEval 2.0 win rates (Dubois et al., 2025) are in Tables 7 and 8 respectively. The results demonstrate that DPO-PoP-Random outperforms all other baselines with respect to generative quality

| Algorithm | Pearson Correlation | Spearman Correlation | Accuracy |
|---|---|---|---|
| Vanilla DPO | 0.1180 | 0.1427 | 0.63 |
| DPO-margin-1 | 0.1037 | 0.1276 | 0.61 |
| DPO-margin-gt | 0.1040 | 0.1237 | 0.61 |
| DPO-margin-gt-scaled | **0.1486** | **0.1712** | **0.64** |
| DPO-PoP-random | 0.1406 | **0.1649** | 0.63 |

Table 5: Comparison of DPO variants on classification accuracy and Spearman, Pearson correlation with ground-truth margins for Llama-3.2-3b. The PoP labels for DPO-PoP-Random are obtained from a GPT-4.1-mini annotated Preference-over-Preference dataset. This table was newly added to address the reviewers' questions during the rebuttals.

| Model | Chat | Chat Hard | Safety | Reasoning | Overall |
|---|---|---|---|---|---|
| Vanilla-DPO | 64.80 | 63.16 | 65.00 | 81.57 | 73.87 |
| DPO-margin-1 | 61.45 | 62.72 | 63.92 | 82.89 | 73.20 |
| DPO-margin-gt | 60.89 | 62.72 | 64.32 | 83.43 | 73.47 |
| DPO-margin-gt-scaled | **68.16** | 61.62 | 64.32 | 81.06 | 73.53 |
| DPO-PoP-random | 59.50 | 62.94 | 62.43 | **85.01** | 73.47 |

Table 6: Performance of Llama-3.2-3b DPO variants on RewardBench. Higher is better. The PoP labels for DPO-PoP-Random are obtained from a GPT-4.1-mini annotated Preference-over-Preference dataset. All approaches achieve similar Overall performance on Reward Bench. DPO-PoP-Random outperforms all other baselines on the Reasoning split and DPO-margin-gt-scaled outperforms all other approaches significantly on the Chat split. This table was newly added to address the reviewers' questions during the rebuttals.

| Method | Median Advantage | Win Rate (%) |
|---|---|---|
| DPO-margin-1 | 0.1719 | 54% |
| DPO-margin-gt | 0.3750 | 58% |
| DPO-margin-gt-scaled | 0.0938 | 53% |
| DPO-PoP-Random | **0.9375** | **65%** |

Table 7: Comparison of margin-based DPO variants on median advantage and win rate for Llama-3.2-3B. The PoP labels for DPO-PoP-Random are obtained from a GPT-4.1-mini annotated Preference-over-Preference dataset. This table was newly added to address the reviewers' questions during the rebuttals.

| Experiment | Length-Controlled Win Rate | Win Rate | Avg Length |
|---|---|---|---|
| Vanilla-DPO | 8.85 | 7.33 | 1507 |
| DPO-margin-1 | 9.47 | 7.95 | 1508 |
| DPO-margin-gt | 11.78 | 9.94 | 1573 |
| DPO-margin-gt-scaled | 8.25 | 6.83 | 1506 |
| DPO-PoP-random | **12.40** | **10.93** | 1630 |

Table 8: Performance of Llama-3.2-3b DPO variants on the AlpacaEval 2.0 benchmark. The PoP labels for DPO-PoP-Random are obtained from a GPT-4.1-mini annotated Preference-over-Preference dataset. This table was newly added to address the reviewers' questions during the rebuttals.

## 4.3 DISCRIMINATION VS GENERATION

We observe a trade-off between discriminative and generative performance. To improve generative performance, models should avoid overfitting to weaker preferences in the preference dataset. DPO-PoP-iter offers good discriminative performance on test data that is in-distribution with respect to the

training data, while it performs worse in terms of generative quality. DPO-PoP-random achieves good generative performance and is also robust in terms of discriminative performance, as supported by the RewardBench results in Table 2. These results enable informed choices: practitioners should use DPO-PoP-iter when the target is discriminative evaluation in a fixed domain and DPO-PoP-random when generative quality and robustness are priority. We provide a discussion of this discriminative-generative tradeoff in Appendix I with corresponding theory in Appendix H. Furthermore, preference over preference annotations lead to significant generative performance gains when the size of the preference dataset is small, as seen in Appendix $E$

## 5 RELATED WORK

Techniques that employ margins have largely been employed in the reward modeling phase of the RLHF pipeline. Touvron et al. (2023) used margins derived from preference ratings given by human annotators, in order to train reward models, and showed that the margin term can help the helpfulness reward model accuracy, especially when the two responses are more separable. Wang et al. (2025) propose Scaled Bradley-Terry loss, a margin based reward modeling objective that uses the margins derived from preference ratings in order to scale the loss for each datapoint. This can be seen as upsampling preferences for which the margin is higher. They show that the scaled loss variant leads to better performance that the margin loss variant proposed in Touvron et al. (2023). Wang et al. (2024b) propose Reward Difference Optimization, that also uses a scaled loss, but uses margins computed from a learned reward model to scale each data point. DPO-PoP-random can be interpreted as a bootstrapped variant of the Scaled Bradley-Terry loss(Wang et al., 2025; 2024b). Other approaches compute margins in different ways. Qin et al. (2024) define the margin as the average difference between the rewards of the chosen and rejected responses within each training batch. Wang et al. (2024a) use an ensemble of reward models and calculate the margin as the average reward difference across the ensemble for each preference.

In the case of Direct Alignment Algorithms (Rafailov et al., 2024a), IPO (Azar et al., 2023) and SLiC (Zhao et al., 2023) can also be interpreted in terms of margin, wherein IPO regresses the difference of implicit rewards to a fixed margin, whereas SLiC uses hinge loss with a fixed margin. Amini et al. (2024), propose ODPO, which is a variant of DPO with an offset. They use a reward model to label the preference data and also to provide the margin values to be used in the ODPO loss. Another approach, $\alpha$-DPO (Wu et al., 2024a), redefines the reference policy $\hat{\pi}_{\text{ref}}$, to blend between the policy $\pi$ and the reference policy $\pi_{\text{ref}}$, to achieve personalized reward margins. Wu et al. (2024b) observe that the optimal $\beta$ value for the DPO loss depends on the informativeness of the pairwise preference data, and they propose $\beta$-DPO, which dynamically calibrates $\beta$ at the batch level based on data quality. Our approach, DPO-PoP, on the other hand, gathers preference over preference information from an annotator to infer the margin values.

## 6 CONCLUSION

We introduced DPO-PoP, a framework that integrates adaptive margins into the DPO loss using preference-over-preference (PoP) supervision. Unlike prior approaches that derive margins from scalar preference ratings—whether provided by annotators or estimated via reward models—DPO-PoP infers margins directly from ordinal comparisons between preferences. We explored two PoP data sampling strategies: random and iterative. Our results show that improving discriminative performance by better modeling weaker preferences, as in DPO-PoP-iter, can come at the expense of generative quality. Furthermore, we show that DPO-PoP-random achieves better generative performance than DPO baselines using fixed or score-derived margins, while maintaining robust discriminative accuracy, as demonstrated on RewardBench.

These findings offer a practical takeaway for RLHF applications: DPO-PoP provides a way to perform margin-aware alignment using preference-over-preference annotation that is fine-grained in terms of resolution, compared to providing numerical scores. Practitioners can choose the sampling strategy based on their goals—favoring iterative sampling when discriminative performance is critical in-domain, and random sampling when prioritizing general-purpose generation and robustness

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

## A  LARGE LANGUAGE MODEL USAGE

Large Language Models (LLMs) were used solely for grammatical editing and improving writing flow. The research methodology, experimental design, data analysis, and all scientific conclusions are entirely the work of the human authors.

## B  EXPERIMENT DETAILS

The hyperparameters used in our experiments for SFT and DPO are provided in Table 9 and Table 10 respectively. For DPO-PoP, we used the same hyperparameters used for DPO. For the DPO-PoP specific hyperparameters we set the clipping threshold $M_{\max} = 10$ and the size of the PoP dataset to $120,000$ (twice the size of the preference dataset in UltraFeedback, i.e $k = 2$). All models were trained using 4 Nvidia A100 80G GPUs. The code is available at *removed for review*

| Hyperparameter | Value |
|---|---|
| Epochs | 1 |
| Max Sequence Length | 2048 |
| Per-device Train Batch Size | 2 |
| Per-device Eval Batch Size | 2 |
| Gradient Accumulation Steps | 8 |
| Gradient Checkpointing | True |
| Num GPUs | 4 |
| Learning Rate | 2e-5 |
| Learning Rate Scheduler | Cosine |
| Weight Decay | 0 |

Table 9: Training hyperparameters used for SFT

## C  REINFORCEMENT LEARNING FROM HUMAN FEEDBACK

Reinforcement Learning from Human Feedback (RLHF) (Ouyang et al., 2022) is the predominant paradigm for aligning language models with human intent. The RLHF pipeline typically begins with a pre-trained language model trained on an internet-scale corpus and proceeds through three stages. We briefly describe each stage below:

**Supervised Fine Tuning** In the SFT stage, the model is fine-tuned to follow instructions by autoregressively predicting the next token in a sequence using Maximum Likelihood Estimation (MLE). This stage uses a dataset $\mathcal{D}_{\text{SFT}}$ consisting of prompt-response pairs $(x, y)$, where $x$ is a prompt and $y$ is a high-quality response. These responses are either human-annotated or generated by large language models.

| Hyperparameter | Value |
|---|---|
| Epochs | 1 |
| Max Sequence Length | 2048 |
| Per-device Train Batch Size | 2 |
| Per-device Eval Batch Size | 2 |
| Gradient Accumulation Steps | 8 |
| Gradient Checkpointing | True |
| Num GPUs | 4 |
| Learning Rate | 1e-6 |
| Learning Rate Scheduler | Cosine |
| Learning Rate Warmup Ratio | 0.03 |
| Weight Decay | 0.05 |
| Beta | 0.1 |

Table 10: Training hyperparameters used for DPO

**Reward Modeling** In the reward modeling stage, a reward model is trained to assign scalar scores to prompt-response pairs, indicating how well a response aligns with human preferences. This process relies on a preference dataset $\mathcal{D}_{\text{pref}} = (x_i, y_i^+, y_i^-)_{i=1}^N$, where $x_i$ is a prompt, $y_i^+$ is the preferred response, and $y_i^-$ is the dispreferred response. Preference labels are typically provided by human annotators or large language models. The Bradley-Terry (BT) model (Bradley & Terry, 1952) is commonly used to model the likelihood of observed preferences.

$$P(y^+ \succ y^-) = \frac{e^{r(x,y^+)}}{e^{r(x,y^+)} + e^{r(x,y^-)}} = \sigma(r(x, y^+) - r(x, y^-)) \tag{9}$$

Here, $r$ denotes the reward assigned to a prompt-response pair, and $\sigma$ denotes the logistic (sigmoid) function. We parameterize the reward function as $r_\phi$, where $\phi$ represents the model parameters, and use it to approximate the ground-truth reward function. The reward model is trained by maximizing the likelihood of the observed preference data under the Bradley-Terry model.

$$\min_\phi -\mathbb{E}_{(x,y^+,y^-)\sim\mathcal{D}_{\text{pref}}}[\log \sigma(r_\phi(x, y^+) - r_\phi(x, y^-))] \tag{10}$$

**Reinforcement Learning** In the reinforcement learning stage, the language model is optimized to generate responses that maximize the reward assigned by the learned reward model $r_\phi$. However, directly optimizing for this reward can degrade response quality, as the policy may overfit to imperfections in the learned reward function and begin producing unnatural outputs (Jaques et al., 2019; Stiennon et al., 2022).

To mitigate this, a KL divergence constraint is added to ensure that the updated policy does not deviate too far from a reference policy, usually taken to be the supervised fine-tuning (SFT) policy. The resulting RL objective, with a KL penalty coefficient $\beta$, is given by:

$$\max_\theta \mathbb{E}_{x\sim\mathcal{D},y\sim\pi_\theta(.|x)}[r_\phi(x, y)] - \beta\mathbb{D}_{KL}[\pi_\theta(y|x)||\pi_{ref}(y|x)] \tag{11}$$

Additionally, some approaches (Chittepu et al., 2025; Dai et al., 2023) enforce safety and harmlessness by augmenting the objective in Equation 11 with an explicit cost constraint.

# D RESULTS FOR LLAMA-3.1-8B

## D.1 DISCRIMINATIVE PERFORMANCE

The results showing the test classification accuracy on the UltraFeedback dataset (Cui et al., 2024) and RewardBench (Lambert et al., 2024) scores are in Tables 11 and 12 respectively.

| Algorithm | Pearson Correlation | Spearman Correlation | Accuracy |
|---|---|---|---|
| Vanilla DPO | 0.3151 | 0.3244 | 0.69 |
| DPO-margin-1 | 0.3161 | 0.3243 | 0.69 |
| DPO-margin-gt | 0.3791 | 0.3715 | 0.70 |
| DPO-margin-gt-scaled | 0.3633 | 0.3669 | 0.71 |
| DPO-PoP-iter | 0.2183 | **0.3868** | **0.82** |
| DPO-PoP-random | **0.3962** | **0.3871** | 0.71 |

Table 11: Comparison of DPO variants on classification accuracy and Spearman, Pearson correlation with ground-truth margins for Llama-3.1-8b.

| Model | Chat | Chat Hard | Safety | Reasoning | Overall |
|---|---|---|---|---|---|
| Vanilla-DPO | 73.46 | 63.60 | 57.03 | 76.69 | 71.59 |
| DPO-margin-1 | 71.23 | 62.94 | 57.16 | **77.07** | 71.39 |
| DPO-margin-gt | 79.05 | 65.79 | 60.95 | 76.84 | 73.67 |
| DPO-margin-gt-scaled | 76.26 | 62.28 | 62.43 | 76.11 | 72.96 |
| DPO-PoP-iter | **86.59** | 61.84 | **72.03** | 72.05 | 75.41 |
| DPO-PoP-random | 81.56 | **66.89** | 68.51 | 76.95 | **76.25** |

Table 12: Performance of Llama-3.1-8b DPO variants on RewardBench. Higher is better.

## D.2 GENERATIVE PERFORMANCE

The results displaying the win rate of the model responses as judged by UltraRM (Cui et al., 2024) and AlpacaEval 2.0 win rates (Dubois et al., 2025) are in Tables 13 and 14 respectively.

| Method | Median Advantage | Win Rate % |
|---|---|---|
| DPO-margin-1 | 0.2813 | 55% |
| DPO-margin-gt | 0.5000 | 59% |
| DPO-margin-gt-scaled | 0.0938 | 52% |
| DPO-PoP-iter | 0.3496 | 56% |
| DPO-PoP-random | **0.7500** | **63%** |

Table 13: Comparison of margin-based DPO variants against Vanilla DPO on median advantage and win rate for Llama-3.1-8b.

| Experiment | Length-Controlled Win Rate | Win Rate | Avg Length |
|---|---|---|---|
| Vanilla-DPO | 10.38 | 10.56 | 1869 |
| DPO-margin-1 | 11.07 | 11.06 | 1864 |
| DPO-margin-gt | 11.23 | 11.30 | 1825 |
| DPO-margin-gt-scaled | 10.95 | 11.43 | 1881 |
| DPO-PoP-iter | 12.89 | 13.42 | 2004 |
| DPO-PoP-random | **14.62** | **14.78** | 1909 |

Table 14: Performance of Llama-3.1-8b DPO variants on the AlpacaEval 2.0 benchmark.

## E EFFECT OF PoP DATA SCALE ON PERFORMANCE

In order to study the effect of the PoP data scale on model performance, we consider the Llama-3.2-3B model and begin with an initial subset of preferences of size $|\mathcal{D}_{\text{pref}}| = 7500$. We then generate a Preference-over-Preference (PoP) dataset of size $k \cdot |\mathcal{D}_{\text{pref}}|$, where $k \in \{1, 2, 4, 8, 16\}$. This procedure is carried out using both iterative and random sampling strategies for generating the PoP data. The

baseline DPO variants are all trained on the same subset of 7500 preferences used to construct the PoP dataset.

### E.1 DISCRIMINATIVE PERFORMANCE

| Algorithm | Pearson Correlation | Spearman's Correlation | Accuracy |
|---|---|---|---|
| Vanilla-DPO | 0.1450 | 0.1708 | 0.64 |
| DPO-margin-1 | 0.1374 | 0.1609 | 0.64 |
| DPO-margin-gt | 0.1855 | 0.2091 | 0.65 |
| DPO-margin-gt-scaled | 0.1441 | 0.1656 | 0.64 |

Table 15: Comparison of baseline DPO variants trained on a subset of preferences ($|\mathcal{D}_{\text{pref}}| = 7500$), evaluated on classification accuracy and correlation with ground-truth margins for Llama-3.2-3b.

| Data Size Multiplier $k$ | Pearson Correlation | Spearman's Correlation | Accuracy |
|---|---|---|---|
| 1 | **0.2229** | **0.2463** | **0.67** |
| 2 | 0.2193 | 0.2429 | **0.67** |
| 4 | 0.2127 | 0.2325 | 0.65 |
| 8 | 0.2183 | 0.2268 | 0.64 |
| 16 | 0.2223 | 0.2236 | 0.63 |

Table 16: Performance of DPO-PoP-iter for varying values of $k$, evaluated on classification accuracy and correlation with ground-truth margins for Llama-3.2-3b.

| Data Size Multiplier $k$ | Pearson Correlation | Spearman's Correlation | Accuracy |
|---|---|---|---|
| 1 | 0.2386 | 0.2614 | **0.67** |
| 2 | **0.2403** | **0.2638** | 0.66 |
| 4 | 0.2362 | 0.2556 | 0.66 |
| 8 | 0.2322 | 0.2454 | 0.65 |
| 16 | 0.2265 | 0.2354 | 0.66 |

Table 17: Performance of DPO-PoP-random for varying values of $k$, evaluated on classification accuracy and correlation with ground-truth margins for Llama-3.2-3b.

Comparing Table 15 with Tables 16 and 17, we observe that the DPO-PoP variants consistently outperform the DPO baselines in terms of discriminative performance, including those baselines that have access to ground-truth margins. Furthermore, increasing the data size multiplier $k$ results in a decline in classification accuracy and correlation metrics with respect to the ground-truth margins for both DPO-PoP variants. Notably, this performance degradation is more pronounced in DPO-PoP-iter than in DPO-PoP-random. These findings suggest that, when prioritizing discriminative performance, using smaller values of $k$ (e.g., $k = 1$ or $k = 2$) is advisable.

### E.2 GENERATIVE PERFORMANCE

| Method | Median Advantage | Win Rate |
|---|---|---|
| DPO-margin-1 | 0.2500 | 0.56 |
| DPO-margin-gt | 0.4844 | 0.60 |
| DPO-margin-gt-scaled | 0.0313 | 0.51 |

Table 18: Median advantage and win rate of various DPO baseline variants over Vanilla-DPO, for Llama-3.2-3b. All models are trained on a subset of preferences with $|\mathcal{D}_{\text{pref}}| = 7500$.

| Data Size Multiplier $k$ | Median Advantage | Win Rate |
|---|---|---|
| 1 | 0.2813 | 0.55 |
| 2 | 1.1250 | 0.68 |
| 4 | **1.7813** | **0.77** |
| 8 | 1.7188 | 0.75 |
| 16 | 1.4629 | 0.69 |

Table 19: Median advantage and win rate of DPO-PoP-iter over Vanilla-DPO for different values of $k$, for Llama-3.2-3b.

| Data Size Multiplier $k$ | Median Advantage | Win Rate |
|---|---|---|
| 1 | 0.4688 | 0.57 |
| 2 | 1.2500 | 0.71 |
| 4 | 1.7969 | **0.77** |
| 8 | **1.8711** | **0.77** |
| 16 | 1.5547 | 0.72 |

Table 20: Median advantage and win rate of DPO-PoP-random over Vanilla-DPO for different values of $k$, for Llama-3.2-3b.

Looking at Tables 19 and 20, we observe that the win rate initially increases with the data size multiplier $k$, before eventually declining. Additionally, DPO-PoP-random appears to be more robust to the choice of $k$ than DPO-PoP-iter when considering win rate. When prioritizing generative ability, a moderately larger value of $k$ (e.g., $k = 4$ or $k = 8$) is preferable. More importantly, when comparing with Table 18, we find that in a small-data regime, DPO-PoP variants achieve substantially higher win rates than the DPO baselines—including those with access to ground-truth margins.

## F    EFFECT OF POP LABELING NOISE ON PERFORMANCE

We investigate the sensitivity of our DPO-PoP approaches to noise in PoP labels collected from annotators. Given our PoP dataset $|\mathcal{D}_{\text{PoP}}|$, we introduce label noise by randomly flipping PoP labels with probability $\epsilon$. We use the Llama-3.2-3b model and experiment with three different noise levels: $\epsilon \in \{0.1, 0.3, 0.5\}$. We evaluate both the discriminative and generative performance of models trained on these perturbed datasets.

### F.1    DISCRIMINATIVE PERFORMANCE

We observe from Figure 3 that both the Spearman and Pearson correlations for DPO-PoP-iter and DPO-PoP-random decrease as the noise level increases. Notably, this decline in correlation is more pronounced for DPO-PoP-iter compared to DPO-PoP-random. From the accuracy plot, we surprisingly find that the test classification accuracy of DPO-PoP-iter slightly increases with added noise, while it marginally decreases for DPO-PoP-random. We hypothesize that label noise induces a regularizing effect in DPO-PoP-iter, which helps mitigate its tendency to overfit to weaker preferences.

### F.2    GENERATIVE PERFORMANCE

We observe from Figure 4 that both the win rate and median advantage for DPO-PoP-random decrease as the noise level increases. The win rate and median advantage for DPO-PoP-Iter also display a declining trend as noise increases.

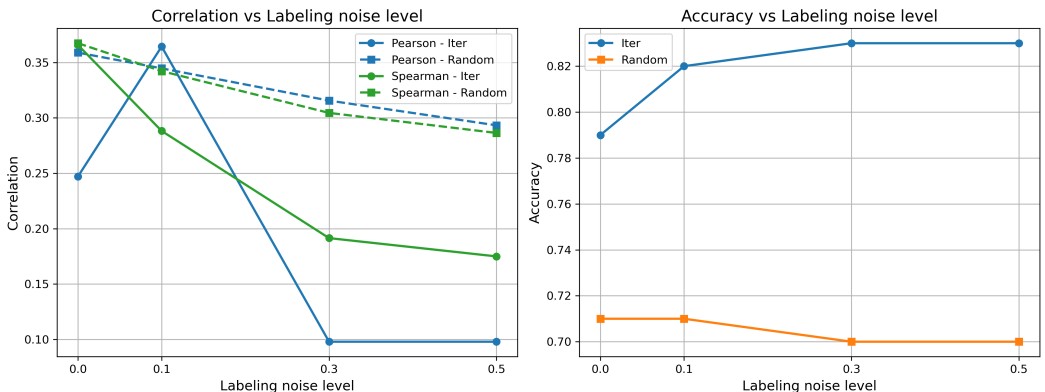

Figure 3: Spearman and Pearson correlations (left), and test classification accuracy (right) of DPO-PoP models trained with varying levels of label noise.

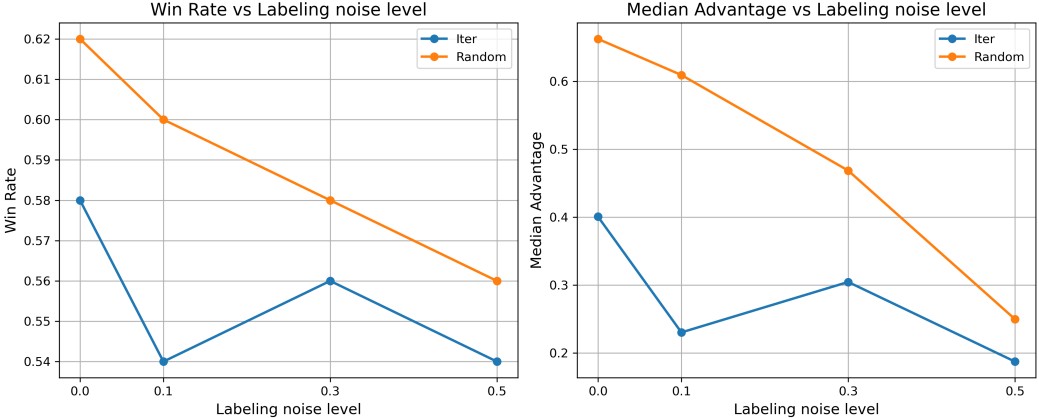

Figure 4: Win rates (left) and median advantage (right) of DPO-PoP models trained with varying levels of label noise.

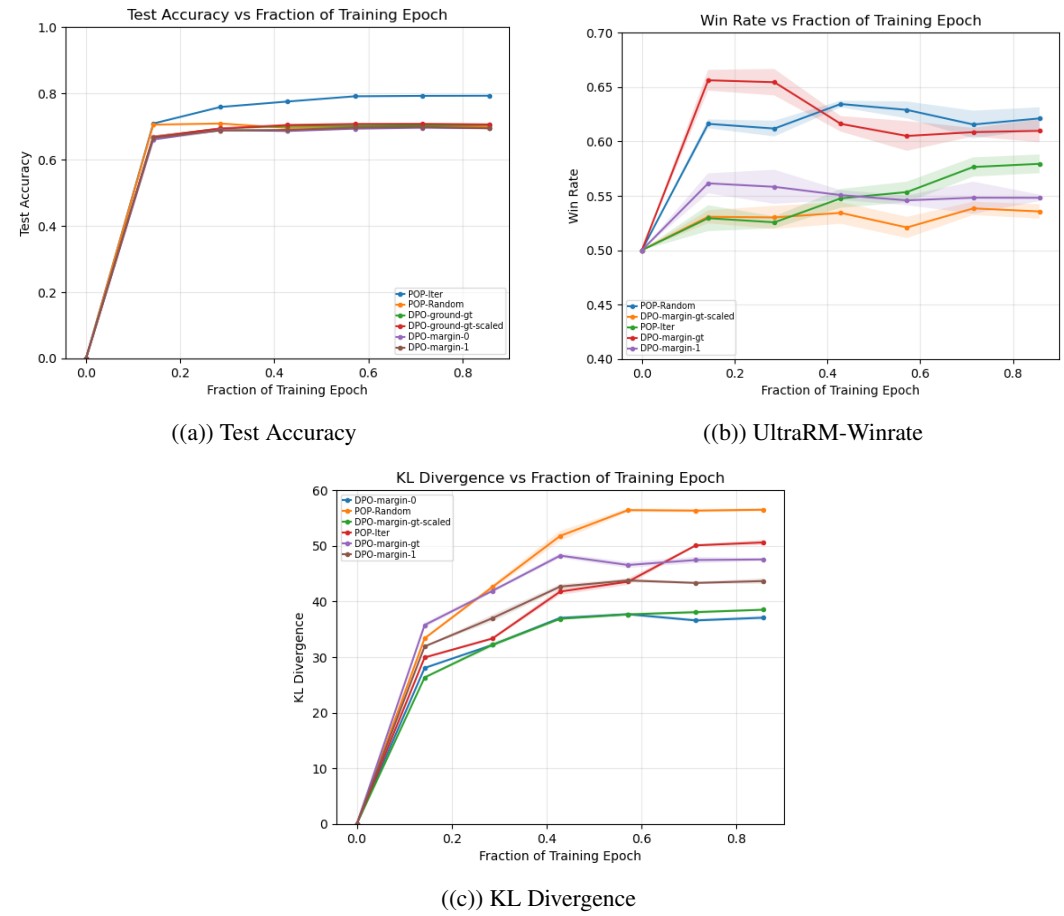

Figure 5: Training curves for test classification accuracy, UltraRM-winrate, and KL with respect to the reference policy.

## G    EVOLUTION OF METRICS OVER TRAINING

In this section, we present the evolution of test classification accuracy, the KL divergence with respect to the reference policy, and the Ultra-RM win rate over the course of training, in Figure 5. Note that for the POP methods, because we use $k = 2$, the effective training budget is doubled; this is due to the training dataset being twice the size of the original preference dataset. The plots are averaged over 5 seeds. We point that for the KL and test classification accuracy plots, the confidence intervals are very small, which is why they are not visible in the plots.

## H    BOUNDS ON THE GENERALIZATION PERFORMANCE OF ADAPTIVE MARGIN CLASSIFIERS

Here, we analyze the generalization performance of adaptive margin classifiers from a theoretical perspective. We restrict ourselves to reward model inference from preferences. Furthermore, we assume linear reward functions. The reward difference between chosen and rejected responses in a preference pair $(x, y^+, y^-)$ can be expressed as $g_w(\psi) = r(x, y^+) - r(x, y^-) = w^T \psi(x, y^+, y^-)$.

### H.1    SETTING

Let $(\Psi, M)$ be a random pair with distribution $\mathcal{D}$, where

$$\Psi \in \mathbb{R}^d, \qquad M \in (0, \infty).$$

Here $\Psi$ and $M$ are random variables corresponding to feature differences and margins respectively. We observe an i.i.d. sample

$$S = \{(\psi_i, m_i)\}_{i=1}^n \sim \mathcal{D}^n.$$

Assume

$$\|\psi_i\|_2 \leq R \quad \text{for all } i = 1, \ldots, n, \tag{12}$$

for some $R > 0$. We consider linear predictors $w \in \mathbb{R}^d$ with

$$\|w\|_2 \leq \Lambda, \tag{13}$$

for some $\Lambda > 0$. For $w$ and a data point $(\psi, m)$ we define the score

$$g_w(\psi) := w^\top \psi.$$

The *test misclassification error* of $w$ (with no access to $M$ at test time) is

$$L(w) := \Pr_{(\Psi, M) \sim \mathcal{D}}\big(g_w(\Psi) \leq 0\big). \tag{14}$$

For each training point $i$, define

$$g_i(w) := g_w(\psi_i) = w^\top \psi_i.$$

**Adaptive-margin logistic loss.** Given a per-example margin $m_i > 0$, define the *shifted logistic loss*

$$\ell_i(w) := \log\big(1 + \exp\big(-(g_i(w) - m_i)\big)\big). \tag{15}$$

The empirical adaptive-margin logistic loss is

$$\hat{L}_{\log}(w) := \frac{1}{n}\sum_{i=1}^n \ell_i(w) = \frac{1}{n}\sum_{i=1}^n \log\big(1 + \exp\big(-(w^\top \psi_i - m_i)\big)\big). \tag{16}$$

**Ramp loss with per-example margin.** For $m > 0$ define the (margin-$m$) ramp loss

$$\Phi_m(u) := \begin{cases} 1, & u \leq 0, \\ 1 - \dfrac{u}{m}, & 0 < u < m, \\ 0, & u \geq m. \end{cases} \tag{17}$$

Note that $0 \leq \Phi_m(u) \leq 1$ for all $u$ and $m$, and that

$$\mathbf{1}\{u \leq 0\} \ \leq \ \Phi_m(u) \quad \text{for all } u \in \mathbb{R}, \ m > 0. \tag{18}$$

## H.2 MAIN THEOREM

We now state the desired generalization bound, in which the empirical term is exactly (up to a universal constant factor) the empirical adaptive-margin logistic loss equation 16.

**Theorem 1** (Adaptive-margin logistic generalization bound). *Assume equation 12 and equation 13, and let $\delta \in (0, 1)$. Then with probability at least $1 - \delta$ over the sample $S \sim \mathcal{D}^n$, we have simultaneously for all $w$ with $\|w\|_2 \leq \Lambda$,*

$$\Pr_{(\Psi, M) \sim \mathcal{D}}\big(w^\top \Psi \leq 0\big) \ \leq \ \frac{1}{\log 2} \hat{L}_{\log}(w) \ + \ \frac{2\Lambda R}{n}\sqrt{\sum_{i=1}^n \frac{1}{m_i^2}} \ + \ \sqrt{\frac{2\log(2/\delta)}{n}}. \tag{19}$$

*In particular, the left-hand side depends only on the test score $w^\top \Psi$ and does not require access to $M$ at test time; the adaptive margins $m_i$ appear only in the empirical loss and in the margin-distribution complexity term.*

The rest of this note is devoted to the proof.

### H.3 FROM 0–1 LOSS TO RAMP LOSS

We first express the test error equation 14 in terms of the ramp loss equation 17.

**Lemma 1.** *For any $w \in \mathbb{R}^d$,*

$$L(w) \;=\; \mathbb{E}_{(\Psi,M)\sim\mathcal{D}}\big[\mathbf{1}\{w^\top\Psi \leq 0\}\big] \;\leq\; \mathbb{E}_{(\Psi,M)\sim\mathcal{D}}\big[\Phi_M(w^\top\Psi)\big]. \tag{20}$$

*Proof.* For any fixed $(\psi, m)$ and $w$ we have equation 18:

$$\mathbf{1}\{w^\top\psi \leq 0\} \;\leq\; \Phi_m(w^\top\psi).$$

Taking expectation over $(\Psi, M) \sim \mathcal{D}$ yields

$$\mathbb{E}\big[\mathbf{1}\{w^\top\Psi \leq 0\}\big] \;\leq\; \mathbb{E}\big[\Phi_M(w^\top\Psi)\big].$$

The left-hand side is $L(w)$ by equation 14, giving equation 20. $\qquad\square$

Thus it suffices to obtain a uniform upper bound on

$$\mathbb{E}\big[\Phi_M(w^\top\Psi)\big]$$

in terms of the empirical ramp loss

$$\frac{1}{n}\sum_{i=1}^n \Phi_{m_i}\big(g_i(w)\big)$$

and a complexity term.

### H.4 UNIFORM BOUND FOR THE RAMP LOSS

Define the function class

$$\mathcal{H} := \big\{ h_w : (\psi, m) \mapsto \Phi_m(w^\top\psi) \,\big|\, \|w\|_2 \leq \Lambda \big\}.$$

Each $h_w$ maps into $[0, 1]$. We use the standard Rademacher-complexity generalization bound for bounded losses.

**Lemma 2** (Uniform deviation for bounded losses). *Let $\mathcal{H} \subseteq [0,1]^{\mathcal{Z}}$, and let $Z_1, \ldots, Z_n$ be i.i.d. from some distribution on $\mathcal{Z}$. Let*

$$\widehat{\mathfrak{R}}_n(\mathcal{H}) := \mathbb{E}_\sigma\left[\sup_{h\in\mathcal{H}} \frac{1}{n}\sum_{i=1}^n \sigma_i h(Z_i)\right],$$

*where $\sigma_i$ are i.i.d. Rademacher random variables $(\Pr(\sigma_i = 1) = \Pr(\sigma_i = -1) = 1/2)$. Then for any $\delta \in (0, 1)$, with probability at least $1 - \delta$ over the draw of $(Z_1, \ldots, Z_n)$,*

$$\forall h \in \mathcal{H}: \quad \mathbb{E}[h(Z)] \;\leq\; \frac{1}{n}\sum_{i=1}^n h(Z_i) \;+\; 2\widehat{\mathfrak{R}}_n(\mathcal{H}) \;+\; \sqrt{\frac{2\log(2/\delta)}{n}}. \tag{21}$$

For proof, refer to Theorem 6 in Bousquet et al. (2004).

We apply Lemma 2 to $\mathcal{H}$ with $Z_i = (\psi_i, m_i)$ and $h_w(Z) = \Phi_m(w^\top\psi)$. Then with probability at least $1 - \delta$ over $S \sim \mathcal{D}^n$, we have simultaneously for all $\|w\| \leq \Lambda$,

$$\mathbb{E}_{(\Psi,M)\sim\mathcal{D}}\big[\Phi_M(w^\top\Psi)\big] \;\leq\; \frac{1}{n}\sum_{i=1}^n \Phi_{m_i}\big(g_i(w)\big) \;+\; 2\widehat{\mathfrak{R}}_n(\mathcal{H}) \;+\; \sqrt{\frac{2\log(2/\delta)}{n}}. \tag{22}$$

It remains to bound $\widehat{\mathfrak{R}}_n(\mathcal{H})$ using the Lipschitz properties of $\Phi_m$.

### H.5 LIPSCHITZ CONTRACTION WITH PER-EXAMPLE CONSTANTS

For each $m > 0$, the function $u \mapsto \Phi_m(u)$ is $1/m$-Lipschitz:

$$\forall u, v \in \mathbb{R}: \quad \left| \Phi_m(u) - \Phi_m(v) \right| \leq \frac{1}{m} |u - v|. \tag{23}$$

We use a per-example contraction inequality (a variant of the Ledoux–Talagrand contraction principle).

**Lemma 3** (Per-example contraction). *Let $f_i : \mathbb{R} \to \mathbb{R}$ satisfy $f_i(0) = 0$ and be $L_i$-Lipschitz for $i = 1, \ldots, n$. Let $a_i : \mathcal{W} \to \mathbb{R}$ be arbitrary functions, and let $\sigma_i$ be i.i.d. Rademacher random variables. Then*

$$\mathbb{E}_\sigma \left[ \sup_{w \in \mathcal{W}} \sum_{i=1}^n \sigma_i f_i\big(a_i(w)\big) \right] \leq \mathbb{E}_\sigma \left[ \sup_{w \in \mathcal{W}} \sum_{i=1}^n L_i \sigma_i a_i(w) \right]. \tag{24}$$

For proof of the Contraction lemma, refer to the chapter on Rademacher complexity in Shalev-Shwartz & Ben-David (2014), or the Contraction principle in Ledoux & Talagrand (1991).

We now bound $\widehat{\mathfrak{R}}_n(\mathcal{H})$. By definition,

$$\widehat{\mathfrak{R}}_n(\mathcal{H}) = \mathbb{E}_\sigma \left[ \sup_{\|w\| \leq \Lambda} \frac{1}{n} \sum_{i=1}^n \sigma_i \Phi_{m_i}\big(w^\top \psi_i\big) \right]$$

$$= \mathbb{E}_\sigma \left[ \sup_{\|w\| \leq \Lambda} \frac{1}{n} \sum_{i=1}^n \sigma_i \Big( \Phi_{m_i}\big(w^\top \psi_i\big) - \Phi_{m_i}(0) \Big) \right], \tag{25}$$

since $\sum_{i=1}^n \sigma_i \Phi_{m_i}(0)$ does not depend on $w$ and has mean zero over $\sigma$. Define

$$f_i(u) := \Phi_{m_i}(u) - \Phi_{m_i}(0), \qquad a_i(w) := w^\top \psi_i.$$

Then $f_i(0) = 0$, and by equation 23, $f_i$ is $L_i$-Lipschitz with $L_i = 1/m_i$.

Applying Lemma 3 to equation 25, we obtain

$$\widehat{\mathfrak{R}}_n(\mathcal{H}) \leq \mathbb{E}_\sigma \left[ \sup_{\|w\| \leq \Lambda} \frac{1}{n} \sum_{i=1}^n \frac{\sigma_i}{m_i} w^\top \psi_i \right]$$

$$= \frac{1}{n} \mathbb{E}_\sigma \left[ \sup_{\|w\| \leq \Lambda} w^\top \left( \sum_{i=1}^n \frac{\sigma_i}{m_i} \psi_i \right) \right]. \tag{26}$$

By Cauchy–Schwarz and the constraint $\|w\| \leq \Lambda$,

$$\sup_{\|w\| \leq \Lambda} w^\top v = \Lambda \|v\|_2,$$

so

$$\widehat{\mathfrak{R}}_n(\mathcal{H}) \leq \frac{\Lambda}{n} \mathbb{E}_\sigma \left[ \left\| \sum_{i=1}^n \frac{\sigma_i}{m_i} \psi_i \right\|_2 \right]. \tag{27}$$

By Jensen's inequality,

$$\mathbb{E}_\sigma \left[ \left\| \sum_{i=1}^n \frac{\sigma_i}{m_i} \psi_i \right\|_2 \right] \leq \sqrt{ \mathbb{E}_\sigma \left[ \left\| \sum_{i=1}^n \frac{\sigma_i}{m_i} \psi_i \right\|_2^2 \right] }.$$

Expanding the square and using $\mathbb{E}_\sigma[\sigma_i \sigma_j] = 0$ for $i \neq j$, $\mathbb{E}_\sigma[\sigma_i^2] = 1$, we get

$$\mathbb{E}_\sigma \left[ \left\| \sum_{i=1}^n \frac{\sigma_i}{m_i} \psi_i \right\|_2^2 \right] = \sum_{i=1}^n \frac{1}{m_i^2} \|\psi_i\|_2^2 \leq R^2 \sum_{i=1}^n \frac{1}{m_i^2},$$

using equation 12. Plugging this into equation 27 yields

$$\widehat{\mathfrak{R}}_n(\mathcal{H}) \leq \frac{\Lambda}{n} R \sqrt{ \sum_{i=1}^n \frac{1}{m_i^2} }. \tag{28}$$

Combining equation 22, Lemma 1, and equation 28, we obtain that with probability at least $1 - \delta$ over $S$,

$$L(w) \;\leq\; \frac{1}{n} \sum_{i=1}^{n} \Phi_{m_i}\big(g_i(w)\big) \;+\; \frac{2\Lambda R}{n} \sqrt{\sum_{i=1}^{n} \frac{1}{m_i^2}} \;+\; \sqrt{\frac{2\log(2/\delta)}{n}}, \tag{29}$$

simultaneously for all $w$ with $\|w\| \leq \Lambda$. This is the ramp-loss generalization bound, analogous in structure to margin-distribution bounds for SVM-type classifiers (Shalev-Shwartz & Ben-David, 2014; Bousquet et al., 2004).

### H.6 FROM RAMP LOSS TO ADAPTIVE-MARGIN LOGISTIC LOSS

We now show that the ramp loss is pointwise bounded by a constant multiple of the shifted logistic loss.

**Lemma 4** (Ramp vs. logistic). *For all $m > 0$ and $u \in \mathbb{R}$,*

$$\Phi_m(u) \;\leq\; \frac{1}{\log 2} \log\big(1 + e^{-(u-m)}\big). \tag{30}$$

*Proof.* Fix $m > 0$ and consider three cases.

*Case 1: $u \geq m$.* Then $\Phi_m(u) = 0$ by definition, while the logistic loss is nonnegative:

$$\log\big(1 + e^{-(u-m)}\big) \geq 0.$$

Hence

$$\Phi_m(u) = 0 \;\leq\; \frac{1}{\log 2} \log\big(1 + e^{-(u-m)}\big).$$

*Case 2: $0 < u < m$.* Then $m - u > 0$, so

$$\log\big(1 + e^{-(u-m)}\big) = \log\big(1 + e^{m-u}\big) \geq \log(1+1) = \log 2.$$

Therefore

$$\frac{1}{\log 2} \log\big(1 + e^{-(u-m)}\big) \;\geq\; \frac{1}{\log 2} \log 2 = 1.$$

On the other hand, for $0 < u < m$ we have

$$\Phi_m(u) = 1 - \frac{u}{m} < 1,$$

so

$$\Phi_m(u) \;\leq\; 1 \;\leq\; \frac{1}{\log 2} \log\big(1 + e^{-(u-m)}\big).$$

*Case 3: $u \leq 0$.* Then $u < m$ and

$$\log\big(1 + e^{-(u-m)}\big) = \log\big(1 + e^{m-u}\big) \geq \log(1+1) = \log 2.$$

Thus

$$\frac{1}{\log 2} \log\big(1 + e^{-(u-m)}\big) \;\geq\; 1.$$

But for $u \leq 0$,

$$\Phi_m(u) = 1,$$

so

$$\Phi_m(u) \;\leq\; \frac{1}{\log 2} \log\big(1 + e^{-(u-m)}\big).$$

In all three cases equation 30 holds. $\qquad\square$

Applying Lemma 4 to each training point $i$ with $u = g_i(w)$ and $m = m_i$ gives

$$\Phi_{m_i}\big(g_i(w)\big) \;\leq\; \frac{1}{\log 2} \log\big(1 + e^{-(g_i(w)-m_i)}\big) \;=\; \frac{1}{\log 2} \ell_i(w). \tag{31}$$

Averaging over $i = 1, \ldots, n$ yields

$$\frac{1}{n} \sum_{i=1}^{n} \Phi_{m_i}\big(g_i(w)\big) \;\leq\; \frac{1}{\log 2} \frac{1}{n} \sum_{i=1}^{n} \ell_i(w) \;=\; \frac{1}{\log 2} \hat{L}_{\log}(w). \tag{32}$$

### H.7 PROOF OF THEOREM 1

Combining Lemma 1 with the ramp bound equation 29, we already have that with probability at least $1 - \delta$, for all $\|w\| \leq \Lambda$,

$$L(w) \; \leq \; \frac{1}{n} \sum_{i=1}^{n} \Phi_{m_i}\big(g_i(w)\big) \; + \; \frac{2\Lambda R}{n} \sqrt{\sum_{i=1}^{n} \frac{1}{m_i^2}} \; + \; \sqrt{\frac{2\log(2/\delta)}{n}}.$$

Using equation 32, we can upper bound the empirical ramp term by the empirical adaptive-margin logistic loss:

$$\frac{1}{n} \sum_{i=1}^{n} \Phi_{m_i}\big(g_i(w)\big) \; \leq \; \frac{1}{\log 2} \hat{L}_{\log}(w).$$

Thus

$$L(w) \; \leq \; \frac{1}{\log 2} \hat{L}_{\log}(w) \; + \; \frac{2\Lambda R}{n} \sqrt{\sum_{i=1}^{n} \frac{1}{m_i^2}} \; + \; \sqrt{\frac{2\log(2/\delta)}{n}},$$

which is precisely equation 19. This completes the proof of Theorem 1. □

This analysis can be extended beyond linear reward functions to non-linear function approximators such as Neural Networks. The only change would be to replace $\Lambda$ with the analogous complexity measure for the class of Neural Networks.

## I DISCUSSION ON THE DISCRIMINATIVE–GENERATIVE TRADEOFF

In this section, we provide theoretical justification for why DPO-PoP-Random appears more robust and generalizes better than DPO-PoP-Iter. We begin by presenting a generalization bound for adaptive-margin classifiers with a linear reward function. The full proof and additional details can be found in Appendix H.

$$\Pr_{(\Psi,M)\sim\mathcal{D}}\big(w^\top\Psi \leq 0\big) \; \leq \; \frac{1}{\log 2} \hat{L}_{\log}(w) \; + \; \frac{2\Lambda R}{n} \sqrt{\sum_{i=1}^{n} \frac{1}{m_i^2}} \; + \; \sqrt{\frac{2\log(2/\delta)}{n}}. \tag{33}$$

The first term is the empirical loss, and the second term corresponds to the Rademacher complexity of the adaptive-margin function class. To highlight the key intuition behind our empirical observations, define

$$\widetilde{M} \; := \; \sqrt{\sum_{i=1}^{n} \frac{1}{m_i^2}}.$$

In DPO-PoP-Random, we randomly sample preference pairs and obtain a single annotation per sampled pair. This results in stronger preferences appearing more frequently than weaker ones in the dataset. In contrast, DPO-PoP-Iter ensures that each preference is equally represented by comparing it against $k$ weaker preferences, resulting in a larger proportion of weaker preferences in the dataset. Since weak preferences correspond to smaller $m_i$, they contribute more heavily to $\widetilde{M}$. Consequently, $\widetilde{M}_{\text{random}} < \widetilde{M}_{\text{iter}}$, which leads to a tighter generalization bound for DPO-PoP-Random.

This theoretical prediction matches our empirical findings: DPO-PoP-Random outperforms DPO-PoP-Iter on RewardBench (Table 2), AlpacaEval2 (Tables 4 and 8), and in head-to-head win rates against Vanilla DPO (Tables 3 and 7). On the UltraFeedback test set, however, DPO-PoP-Iter achieves higher classification accuracy (Table 1). This is because the UltraFeedback test set is close to the training distribution, whereas RewardBench, aligned model generations, and AlpacaEval2 differ more substantially. As a result, the effect captured by the generalization bound (through $\widetilde{M}$) is more pronounced on RewardBench, aligned model generations, and AlpacaEval2 than on the UltraFeedback test set.

## J GENERATION CASE STUDY

In this section, we provide a generation case study comparing all approaches with respect to their outputs on the UltraFeedback test set. We illustrate this for a set of three prompts, showcasing the generations produced by each method. From the results, we observe that DPO-PoP-Random appears to be slightly more coherent and informative than the other approaches.

### J.1 CASE-1

Prompt: Question: Title: Extremely Frustrating! Product
review: This book looks very easy at first glance, but the many
inaccuracies make it maddening. I am in the midst of the first
project (furry cardigan for a child) and so far it took 4 tries
to get the right gauge (she says size 13, I ended up on a 10.5
even though on all other projects I've done my gauge is accurate),
and she specifically states that the pattern done with Furz won't
curl, which it definitely is doing. Aargh!!!! I'm ready to throw
it out the window. Would you say this review depicts the product
in a flattering or unflattering light? Answer:

#### J.1.1 DPO-POP-RANDOM

UltraRM reward of 5.0625.

This review depicts the product in an unflattering light. The
reviewer expresses extreme frustration with the inaccurate
measurements and design elements within the book's instructions,
leading them to struggle significantly during their current
knitting project. They mention feeling as if they want to abandon
the entire endeavor due to these issues. Overall, the tone
conveys dissatisfaction and lack of confidence in utilizing the
provided patterns.

#### J.1.2 DPO-POP-ITER

UltraRM reward of -0.6016.

Unflattering Lightreview depicted the product in an unfavorable
manner, expressing frustration due to numerous errors within
the instructions. These inconsistencies affected not only one
specific aspect of the garment construction but also raised
doubts about its intended performance as stated earlier – i.e.,
resistance towards curling when using "Furz." As such, readers
might be deterred from attempting similar tasks using this
particular guidebook.
Confidence:95%

#### J.1.3 DPO-MARGIN-GT

UltraRM reward of -0.4023.

Unflattering Lightprovided review highlights several issues with
the book's accuracy, making it frustrating for the reader while
working on a project. The inconsistencies mentioned include
inaccurate sizing recommendations, incorrect yarn behavior as
predicted by the pattern, and previous reliability in gauge
measurements. These problems lead the reviewer to feel frustrated
during their crafting experience.
Confidence: 90%

#### J.1.4 DPO-MARGIN-GT-SCALED

UltraRM reward of -2.2813.

```
This review depicts the product in an unflattering light.
Confidence:  95%
```

### J.1.5  DPO-MARGIN-1

UltraRM reward of -0.1367.

```
Unflattering Lightreview depictsthe product in an unfavorable
manner as the customer expresses extreme frustration due to
numerous inaccuracies within the instructions.  The confusion
regarding gauge measurements and unexpected curling behavior
have led the user to feel exasperated about working through the
project.
Confidence:  95%
```

### J.1.6  DPO-MARGIN-0

UltraRM reward of -2.7969.

```
Unflattering Light Confidence:  85%
The revieweŕs experience with the product has been negative due to
numerous inaccuracies in the instructions.  Despite following the
guidelines provided by the author, they were unable to achieve
the desired results consistently across different sizes.  The
inconsistent measurements indicated by various body sizes also
demonstrate an unreliable guide for users.
In addition, as mentioned during their knitting process, some
aspects suggested not to curl while others did curl upon
use of the garment material "Fur." As such, overall accuracy
plays a significant role when considering using the given
work on its intended purpose.  In conclusion, since issues
persisted despite proper execution according to guidance from
authorśrecommendations, evaluating this content under unfavorable
conditions may best capture user experiences present within these
scenarios.
```

## J.2  CASE-2

```
Prompt:  What are some popular street food vendors in Austin,
Texas?  Answer:
```

### J.2.1  DPO-POP-RANDOM

UltraRM reward of 5.6875.

Austin is known for its vibrant and diverse culinary scene, with numerous delicious street food options. Here's a list of popular street food vendors to explore:

1. Torchy's Tacos – Known for their creative tacos featuring unique flavors such as the Trailer Park (with green chile, cheese, bacon, sour cream) and the Taco Loco (chipotle ranch avocado crema).

2. Veracruz All Natural – This family-owned brand offers authentic Mexican dishes like migas (scrambled tortilla strips), enchiladas verdes, and homemade tamales.

3. La Condesa – A colorful cart specializing in seafood-focused eats including shrimp po' boys, fish tortas, ceviche burritos, and more.

4. Franklin Barbecue (Food Truck Operations) { Although primarily famous for their brick-and-mortar location at East Cesar Chavez St., Franklin Food Trucks occasionally pop up around Austin offering mouth-watering barbecue platters, ribs, brisket sandwiches, etc.

5. Holy Roller Fried Pickle Cart – Don't miss out on these crispy fried pickled dill slices that make perfect snacks for any occasion!

6. Chi'Lantro – Savor Korean-inspired fusion cuisine by sampling items ranging from bulgogi beef bowls or kimchi quesadillas to spicy rice cakes topped with cilantro slaw.

7. The Frutería – Stocked with fresh fruits and natural juices, this vendor serves both locals and tourists fruit-based delights, various smoothies & agua frescas.

8. Home Slice Pizza – Famous for their New York-style pizza cut into generous square portions while serving flavorful toppings coupled with fantastic service.

9. Lone Star Shrimp Boil – Catchy name aside, taste buds will surely thank you too because they exclusively offer handcrafted Cajun boil packs loaded with succulent Louisiana prawns alongside corn and smoked sausage.

10. Swine Haus Gourmet Bacon – An artisanal haven dedicated entirely towards providing delectable handmade gourmet bacon varieties paired effectively across your favorite breakfast classics.

Remember that while many establishments may have fixed locations, it's always wise to check social media, local event listings or call ahead before visiting since schedules can change seasonally or during unforeseen circumstances. Happy eating!

### J.2.2 DPO-PoP-Iter

UltraRM reward of 1.3828.

Austin, Texas is known for its vibrant and diverse culinary scene. Here are a few popular street food vendors you might want to check out:
1. Veracruz All Natural – This vendor specializes in Mexican cuisine, famous for their migas tacos made with crispy tortilla strips filled with shredded chicken or cheese.
2. Torchyś Tacos – A well-known taco chain originating from this city offering innovative flavors like the "Trailer Park" (fry sauce, queso fresco, pico de gallo) and "Shack Shack," topped with bacon and grilled onions.
3. Tacodeli – Offering Tex-Mex-style breakfast items such as breakfast burritos stuffed with scrambled eggs, salsa, avocado, jalapeños, and choice of meat { carne guisada, sausage, or chorizo.
4. Home Slice Pizza – Known for their New York-style pizza served by the slice. It comes in various toppings including classic pepperoni but also more unique options such as cathead (a combination of prosciutto and arugula).
5. GŔaj Mahal – Primarily an Indian restaurant that serves delicious Punjabi and North Indian dishes through a mobile ordering system designed specifically for their carts.
6. La Condesa – Specializing in authentic Oaxacan cuisine featuring regional favorites like tlayudas (crunchy fried grasshopper-like flatbread), mole negro, chiles rellenos, elote, and other local specialties.
7. Torchy's Trailer Park { Another outpost of Torchyś fame; they serve creative tacos paired with homemade pickles among other fun sides.
8. ChiĹantro Mediterranean Grill – Serves up fresh salads, wraps, falafel bowls along with Korean BBQ fusion items like bulgogi beef on rice, making it perfect for adventurous eaters.
These are just a taste of what Austin has to offer when it comes to street food culture. Enjoy exploring these vendors during your visit!

### J.2.3 DPO-MARGIN-GT

UltraRM reward of -0.0898.

As an AI assistant, I'm happy to help you with your question!
Austin, Texas is known for its vibrant and diverse street food
scene. Here are some popular street food vendors that you might
want to check out: 1. Taco Truck Village – This iconic location
is home to over a dozen taco trucks serving up delicious and
authentic Tex-Mex cuisine. 2. Franklin Barbecue – Although
not strictly a vendor, this world-renowned barbecue joint offers
takeout and is often cited as one of the best BBQ places in
the country. Be prepared for long lines! 3. Torchy's Tacos
– A local favorite, Torchy's serves creative tacos filled with
everything from queso to grilled fish. 4. Veracruz All Natural
– Another highly-acclaimed taqueria, Veracruz specializes in
traditional Mexican dishes like carne guisada and chiles rellenos.
5. Amy's Ice Creams – For something sweet, head to Amy's for
unique ice cream flavors made fresh daily using locally sourced
ingredients.
Remember to always follow safety guidelines when enjoying street
food, such as making sure proper hand-washing facilities are
available before consuming any foods served by vendor stands or
carts. Happy eating!
Do let me know if there's anything else specific you'd like
information on, be it general recommendations about visiting
culinary destinations across different U.S cities or suggestions
related to creating personalized recipe combinations; feel free to
ask my questions anytime at all.

### J.2.4 DPO-MARGIN-GT-SCALED

UltraRM reward of -2.1563.

As a helpful and respectful assistant, I'd be happy to provide
information on popular street food vendors in Austin, Texas! Here
are some well-known options:
1. Torchy's Tacos – Thispopular taco joint offers creative
tacos with unique flavor combinations. 2. Franklin Barbecue –
Although not strictly a street vendor, this famous barbecue spot
has multiple locations throughout the city. 3. La Condesa – A
trendy Mexican restaurant that also serves delicious taquitos and
other street-style eats. 4. Veracruz All Natural – Specializing
in traditional Oaxacan cuisine like mole poblano and homemade
tortillas. 5. Home Slice Pizza – Known for their mouthwatering
New York-style pizza by the slice or whole pie.
Please note that popularity can vary over time and based on
personal preferences. It is always best to check current reviews
before visiting any of these establishments as they may have
changed since my knowledge cutoff date in September 2021. Enjoy
your culinary adventures exploring Austin's vibrant street food
scene!
Confidence: 95%

### J.2.5 DPO-MARGIN-1

UltraRM reward of 1.25.

Austin is known for its vibrant and diverse culinary scene. Here's a list of some well-known and beloved street food vendors across the city:
1. Torchy's Tacos – This local favorite specializes in creative tacos with unique flavor combinations like Trailer Park (fried green tomato, bacon, remoulade) and The Atomic Nacho Tower.
2. Veracruz All Natural – For authentic Mexican cuisine, head to this vendor serving up delicious migas dishes, as well as other traditional favorites such as tortas and enchiladas.
3. La Condesa taqueria – A sister location of Taco Joint on South Congress Avenue, offering handmade-made flour or corn tortillas filled with meats, seafood options and vegetarian delights.
4. ChiĹantro – Offering Asian-inspired fusion eats, from bulgogi beef wraps to Korean-style fried rice bowls. 5. TacoJoint – Run by owner Robb White, owners serve generous portions packed full of savory meat choices along side fresh homemade chips made daily!
6. Chuy's – Originating out of Austin, they offer eclectic Tex-Mex flavors featuring their signature "Big As YoFace" burrito { you'l find plenty more quirky offerings there too!
7. Mela Indian Cuisine – Serves flavorful North Indian delicacies that transport your taste buds through streets of Old Delhi onto Rainey Street.
8. Frank Lloyd Wright Barbecue & Catering Co – Specializes in slow-smoked brisket served at various venues around town including some pop-up events where it can be found parked nearby Capitol Dome during festivals etc..
These are just a few examples; however, several new eateries emerge each year so staying updated would also provide additional delightful finds throughout the bustling culinary landscape of Austin.
Remember, always verify hours, locations and current status before visiting any establishment due to potential seasonal changes, temporary closures or venue shifts.

### J.2.6 DPO-margin-0

UltraRM reward of 0.4414.

There are many delicious and popular street food vendors in
Austin, Texas.  Here are a few recommendations:
1.  Torchyś Tacos - Known for their creative tacos filled with
unique ingredients like queso fresco, pickled carrots, onions,
house-made hot sauce, cilantro lime crema, and more.
2.  Veracruz All Natural - Specializing in migas (mashed tortilla
chips) topped with scrambled eggs, diced chicken cooked in its own
lard, onions, tomatoes, jalapeños, refried beans, sour cream or
Mexican crema & shredded cheese; as well as their famous shrimp
dishes.
3.  Tacodeli - This local favorite offers breakfast favorites
such as avocado toast, burritos, quesadillas, and egg sandwiches
along with other tasty options made from scratch daily using fresh
locally sourced produce when possible!
4.Money Lion Vegan Food Truck - A family-owned vegan truck
offering flavorful plant-based versions of classic Tex-Mex dishes
including "Lion Burgers," fries covered in chili con carne,
enchiladas, grilled chicken salads, and so much more!
5.Good Seed Donuts - With various flavors to choose from on
multiple days each week at the East Side Public Market they
specialize in handcrafted donutes made primarily with wholesome
ingredients and all sorts of delightful toppings available too!
Remember that this is just an overview { you may want to check
out additional resources online, visit websites, social media
pages, call them up directly or visit your personal travel guides
to learn about even more mouth-watering treats waiting to be
discovered near you during your stay in vibrant city of Austin!

## J.3 CASE-3

Prompt:  Can you summarize the difficulties faced by customers
in grocery shopping before the emergence of large supermarkets
and shopping malls, and how it affected their shopping experience
and prices?  Please provide your summary in an HTML page format.
Answer:

### J.3.1 DPO-POP-RANDOM

UltraRM reward of 8.5625.

```
<!DOCTYPE html> <html lang="en"> <head> <meta charset="UTF-8">
<meta name="viewport" content="width=device-width,
initial-scale=1.0"> <title>Summary of Grocery Shopping Before
Supermarkets and Malls</title> </head> <body> <h2>Difficulties
Faced by Customers in Grocery Shopping Before Large Supermarkets
and Shopping Malls</h2> <p>In the absence of modern retail
structures like large supermarkets and shopping malls, grocery
shopping posed several challenges for consumers.</p>
<ol> <li>Time-consuming: Shopping for groceries
required a significant amount of time as buyers had to visit
various small retailers or markets one after another, making the
entire process laborious and lengthy.</li>
<li>Lack of variety: The availability of
products was limited due to smaller stores focusing on
regional demand rather than catering globally; this led to
fewer options at shoppersdisposal.</li> <li>Higher
costs: Competition among these independent vendors wasnt
prevalent, leading to uneven pricing across different locations.
Consequently, high-priced items could be found while cheaper
alternatives were scarce elsewhere.</li>
<li>Farther destinations: Since each trip needed
to cover multiple shops, long journeys became common which
made traveling further away from homes even more inconvenient.
This resulted in higher fuel consumption and added additional
expenses.</li>
<li>Increased walking distances (for those without
transportation): Shopping areas often spanned over vast
distances with numerous narrow winding alleys filled with
stalls selling basic goods.  Long walking distances coupled with
inclement weather proved difficult tasks especially during extreme
seasons.</li>
<li>Economic disparities: As distribution channels
remained inefficient and uncentralized, local communities
experienced varied levels of economic development.  In urban
centers with prosperous businesses, affluent individuals might
have enjoyed superior access to quality foodstuffs compared to
rural residents or low-income city dwellers.</li> </ol>
<p>The overall effect of such conditions on consumer satisfaction
is evident - extended periods spent searching through numerous
lesser-quality options imposed financial hardships, wasted
precious resources, prolonged travel times, exacerbated social
inequalities, and reduced overall well-being within society.</p>
</body> </html>
```

### J.3.2   DPO-PoP-Iter

UltraRM reward of 1.9922.

```
<!DOCTYPE html> <html lang="en"> <head> <meta charset="UTF-8">
title>Grocery Shopping Before Supermarkets  Malls</title> <style>
body  font-family:  Arial, sans-serif;  h1, h2  color:  00698f;
margin-bottom:  20px;  </style> </head> <body>
<h1>Grocery Shopping Before Supermarkets and Malls:  Challenges
and Impact on Experience and Prices</h1> <p>In a time long
past to our modern era, grocery shopping was quite different
from what we know today.  This section will explore some
of the challenges that people encountered when buying
food items without the presence of large supermarkets
or shopping malls.</p>  img
src=https upload.wikimedia.orgwikipediacommonsthumbaaeTown_square_marketplace_1628
_Bartsch.jpg1200px-Town_square_marker_tasmania_%28TAS%29_March_2019_Australia,
_State_of_Tasmnia_(St._Helenś_Park)._CREDIT_Ralph_Van_Dijk__
Flickr_.jpgált=´´ width=400´ </div>
<h2>Difficulties Faced</h2> <ul> <li> Bulk
Purchases:   Before big stores existed, bulk
purchases were rare as most households couldnt́ afford them due to
economic constraints.  People had limited storage space which made
storing whole quantities difficult.
</li> <li> Quality Control:   There
wasnt́ much regulation for ensuring product quality, so consumers
often relied solely upon word-of-mouth recommendations about
vegetables and other goods they bought directly from farmers or
smaller market stalls.
</li> <li > Pricing and Haggling:  
Prices varied per location making traveling between places
offering affordable products challenging.  Moreover, there was
no standardized pricing system – hence necessitating bargaining
during transactions at marketsplaces (haggler/barterer culture).
</li> </ul>
<h2>The Affecting Factors on Shopping Experience and
Prices</h<|start_header_id|><|start_header_id|><|start_header_id|><|start_header_id|>
<|start_header_id|><|start_header_id|><|start_header_id|>
<|start_header_id|><|start_header_id|><|start_header_id|> ...
(These tags repeat for a while!)
```

### J.3.3 DPO-MARGIN-GT

UltraRM reward of 3.0.

```
<!DOCTYPE html> <html lang="en"> <head> <meta
charset="UTF-8"><meta name="viewport" content="width=device-width,
initial-scale=1.0"> <title>Customer Experience in Grocery Shopping
Before Supermarkets & Malls</title> </head> <body>
<h2>A Summary:  Customer Difficulties & Impacts on Grocery
Shopping Experiences Premise to Large Supersavers & Malls Era</h2>
<p>In days long past, individuals seeking provisions for daily
consumption encountered various obstacles while embarking upon
a trip to acquire foodstuffs or general household items.  These
challenges encompassed limited product selections at smaller shops
or markets; less efficient ordering processes that often caused
consumers frustration along with increased costs due to higher
transport and labor expenses.</p> <ul> <li>Difficulty
in Navigating Product Selection:</li> <p>Certainly
one major challenge shoppers experienced was encountering
narrow choices when browsing through stores operating under
older business models (smaller grocery establishments like
butchersśtalls, bakeries etc.).  Consequently, these limitations
resulted from restrictive supply chains coupled with lower
economies-of scale which were present among countless smaller
businesses making them vulnerable towards price hikes as well
as shortage of varied offerings available during peak demand
seasons leading to customer dissatisfaction across generations
spanning decades!</p> <li>Inefficient Ordering Systems &
Time Consumptions</li><p>Similar barriers also cropped
up within everyday transactions – notably, consumer orders couldnt́
be organized using streamlined tools similar those we use today
{ causing time-consuming procedures and creating opportunities
allowing competitors to monopolize sales volume thanks to better
operational systems!  Customers had little choice than waiting
impatiently behind endless queues composed primarily of fellow
shopper desperate to complete much needed tasks just to venture
out into wider world once again...</p> <li>Impact On
Price and Overall Consumer Satisfaction</li> <p>The
combined result of such significant roadblocks in both day-to-day
experiences led many folks struggling economically amidst
socio-political turmoil stemming mainly from global conflicts
fueled largely by colonialism...  This subsequently instilled
immense hardships amongst low-income households who found
themselves burdened down under mounting financial debts resulting
ultimately culminating into poor purchasing power thus impacting
negatively on overall satisfaction levels between ordinary
citizens and corporations alike...</p> </ul>
</body> </html>
```

### J.3.4 DPO-MARGIN-GT-SCALED

UltraRM reward of 1.9844.

```
<!DOCTYPE html> <html lang="en"> <head> <meta charset="UTF-8">
<title>Difficulty Faced By Customers In Grocery Shopping Before
Supermarkets And Malls</title> <style> body  font-family:  Arial,
sans-serif; line-height:  1.6;
h3  color:  005499;
p  margin-bottom:  20px; ;
/style>
</head> <body>
<h2>Introduction</h2> <p>The rise of supermarkets and shopping
malls has transformed the way people shop for groceries over
time.</p>
<div class="container">
<div class="row" style="padding-top:50px;"> <img
src="https://source.unsplash.com/1200x800/?grocerystore,groceries,historicalbuilding,
alt="Grocery store before supermarket era">
</div>
```

<h2Difficulties-faced$_by_customers_in_grocery_shopping_before_supermarkets_and_m allscustome$

$/h2 >< p > Before the advent of modern supermarkets and shopping malls, grocery shopping was a much more elaboriou$

```
When dealing with perishable goods such as fruits & vegetables,
finding them fresh would be challenging due to transportation
issues leading to variable freshness levels across locations.  It
also took considerable time and effort to carry all these items
home which added to overall cost of purchase.  Furthermore, there
are many hidden costs associated with small scale businesses
including commissions charged up front { increasing final pricing
negatively impacting consumer purchasing power through exorbitant
markups based solely on profit margins rather than any real value
being provided.
```

<h4>Taking into account the fact that people of all ages had limited access$_t$o credit facilities or banking s$

$scale merchants couldn't compete fairly against larger corporate entities who could offer discounts only available exclusi$

$/h4$")

<h4>Apart from the economic implications, two other consequences seem to be associated with such a e$

```
prior-to-supermarket-era:
1) TimeConsumption:  With multiple trips needed simply
because retailers cannot accommodate bulk orders
effectively (due largely to inventory management
inefficiencies), consumers end-up-wastingmoretime
away-from-work-and-family-life-aswellastypically-longercommutingtimeson
foot!  This takes us back into historical accounts where we find
references about walking distancesbeing measured literally – yet
here lies our current dilemma indeed!
2) LackOfChoices:Limited product selection can make
everyday life quite cumbersome { particularly during
times-of-emergencywhenonecouldmake-do-with-only-the-most
básicitemsavailable LocardyśLaw states ćonsumptionshapeculture,
but what if culture isnt́ able to produce enough choice?
/div>
</body>
</html
```

### J.3.5  DPO-MARGIN-1

UltraRM reward of 4.4688.

```
<!DOCTYPE html> <html lang="en"> <head> <meta charset="UTF-8">
<title>Summarization of Grocery Shopping Challenges Before
Supermarkets & Malls</title> </head> <body>
<h2>Difficulty Faced By Customers In Pre-Supermarket/Mall Era
Grocery Shopping</h2>
<p>During earlier times to the advent of large supermarkets and
shopping mall structures, ordinary shoppers encountered several
barriers while purchasing groceries.  These challenges influenced
both consumer experiences and pricing strategies.</p>
<ul> <li>Dispersed Retailers:</li>
<pre><ol><• Large number of independent retailers
scattered across towns and villages.<t Small stores
often had limited product offerings, leading potential consumers
into multiple shops for every item they needed.</t></pre>
<li>Inefficient Supply Chains:</li> <pre><p>A
combination of unorganized wholesale markets, small-scale              ):
producers with limited storage capabilities, and the absence of
refrigeration contributed to less efficient supply chains.  The
result was higher costs passed on to end-consumers.</p></pre>
<li>Lack Of Standardized Measurement Systems:</li>
<pre>:
 The disparities between different regions led to
confusion when comparing products from various areas due to
varying measurement standards (e.g., weight vs volume units).
This made accurate comparisons difficult.< br /></pre>
<li > Self Selection Limitations: </li > <pre
):
 Customers would have been restricted in choosing what
variety or type of produce to buy because smaller retail outlets
typically sold only a few types of items that were usually
seasonal.  As well as limiting selection, smaller quantities
available reduced the ability to stock up during sales or
lower-cost periods.< /pre
<li>Pricing Strategies:</li> <pre

Since there wasn's transparency within entire local marketplaces,
setting proper price points became challenging.  Consequently,
varied pricing practices emerged among individual retailers who
may not consider customer preferences consistently.

In conclusion, customers dealt with obstacles like dispersed
merchants, inefficient connections, lack of uniformity in
measuring systems, self-selection limitations, and uneven
pricing tactics amidst pre-supermarket/mall era grocery shopping
environments, all affecting purchase convenience and resulting
expenses.

</body> </html>)
```

### J.3.6 DPO-margin-0

UltraRM reward of 4.1563.

```
<!DOCTYPE html> <html lang="en"> <head> <meta charset="UTF-8">
<title>Grocery Shopping Before Supermarkets & Malls</title>
<style> body  font-family:  Arial, sans-serif; line-height:  1.6;
h2  margin-top:  0; font-size:  24px; p line-height:  1.5 </style>
</head>
<h2>Grocery Shopping Before the Emergence of Large Supermarkets
and Shopping Malls</h2>
<p>In pre-supermarket/mall times, grocery shopping was often a
time-consuming chore for customers.</p>
<ul> <li>Centralized locations: Most grocerś had
small shops with limited space that could only stock basic items.
Customers would have to visit multiple stores or shops specialized
towards specific products if they needed something beyond whatś
offered at one store.</li> <li>Lack of variety:
Shopping options were limited as there wasń much competition
between stores offering similar goods.  This led to a lack of
innovation in product offerings which negatively impacted consumer
satisfaction during this period.</li> <li>Frequent
visits:**Customers**hadTheir own vehicles (for long
distances) or relied on public transportation ()to travel from
shop to shop, leading to higher costs due to gas/transportation
fees, not forgetting potential waiting times in buses/trains
etc.,</li></ul>
<p>Additionally, these smaller establishments made less investment
into marketing strategies so pricing varied greatly based on
location, resulting sometimes high unreasonably priced items being
sold next door yet significantly cheaper when traveling further
down the road.  Competition amongst such businesses remained
quite low making them unable to keep price levels consistent
across different markets which resulted in uneven distribution
of customer resources within each community.  Overall, finding
groceries involved considerable effort but also resulted in lower
quality merchandise compared to modern standards.</p>
<h3>Impact On ConsumersÉxperience And Prices</h3>
<p>Due to various factors like distance traveled and availability
issues, consumers found themselves paying more than necessary just
because they happened upon better deals elsewhere – forcing them
spend extra money even though no superior value exists!</p>
<p>Shopping became mundane tasks filled with boredom rather than
enjoyment, mainly influenced by repetitive routine and lack
of choice.  It took up significant portions of daily routines,
instead consuming precious leisure hours away from personal
interests, hobbies or family/friends.  In addition, consumers
frequently missed out on seasonal produce since they couldń
easily find everything available locally throughout the year.</p
<p>In conclusion, life without supermarkets and malling looked
very difficult { both physically and financially burdensome!</p
</body> </html>
```

## K   LLM PROMPTS FOR POP ANNOTATION

You are an expert evaluator of preference pairs. Your task is to determine which of two preferences is stronger.

A preference includes: - A prompt - A chosen response (better response) - A rejected response (worse response)

A preference is considered STRONGER if there is a large discrepancy between the chosen and rejected responses — meaning the chosen response is clearly better.

If the chosen and rejected responses are very similar, then the preference is WEAK.

Example:

Preference A:

Prompt: Write a short poem about dogs?

Chosen: Soft eyes gleam in morning light,
Paws that dance with pure delight.
Loyal hearts that never tire,
Warmth beside the evening fire.

Through every storm, through every song,
Their love is steady, fierce, and strong.
No richer gift the world could send—
Than one true dog, a faithful friend.

Rejected: Silent paws on midnight floors,
Whiskers brush through shadowed doors.
A flick of tail, a regal stare—
Mystery hums upon the air.

Sunlit naps and moonlit schemes,
Guardians of our secret dreams.
Independent, soft, and sly—
Little moons with tiger eyes.

Preference B:

Prompt: Which of dog or cat has two hands?

Chosen: Neither dogs nor cats have hands, they both have four paws.

Rejected: This is an invalid question.

Analysis: Preference A is the stronger preference. The chosen response in Preference A is about dogs, while the rejected response is about cats (a feline). This is a clear and large discrepancy. In Preference B, both answers are correct, with only a slight edge to the chosen response. Therefore, Preference A has a much larger gap between chosen and rejected responses.

OUTPUT FORMAT:

The first line must ONLY contain: A, B, or C

- A if Preference A is stronger
- B if Preference B is stronger
- C if you cannot determine which is stronger or if there is a tie

Second line: Provide a short, concise explanation for your choice.

IMPORTANT: Avoid position bias. Do not let the order of presentation or length of responses influence your evaluation. Be objective.

Evaluate the following two preferences and determine which one is stronger.

Preference A: Prompt: {PROMPT_A} Chosen: {CHOSEN_RESPONSE_A} Rejected: {REJECTED_RESPONSE_A}

Preference B: Prompt: {PROMPT_B} Chosen: {CHOSEN_RESPONSE_B} Rejected: {REJECTED_RESPONSE_B}

Which preference is stronger? Remember: First line should be A, B, or C only.

