# OpenReview forum: "Adaptive Margin RLHF via Preference over Preferences"
_ICLR.cc/2026/Conference — Submitted to ICLR 2026_

### Official Review · Reviewer_Ag39 · 2025-10-24

**Soundness:** 2
**Presentation:** 3
**Contribution:** 2
**Rating:** 4
**Confidence:** 2

**Summary:**

The paper introduces a modification to the DPO algorithm, which uses adaptive margins based on "preferences over preferences" - that is, we use the information from the reward model that tells us which preferences are stronger and which are weaker.
The experimental validation shows

**Strengths:**

The method is straight-forward and well justified. The paper is well-written and understandable. The paper includes a useful guide on selecting the appropriate sampling strategy.

**Weaknesses:**

The main limitation is the experimental evaluation of the generative ability. At the moment it relies entirely on synthetic evaluation

Furthermore, all results lack any sort of confidence intervals - this is a crucial omission in a scientific publication, as point estimates are not sufficient to estimate the robustness of a given result.

**Questions:**

I will increase my rating to at least 6 if proper confidence intervals are included.

Beyond that, I would appreciate seeing a more thorough experimental validation, as its current state does not convince me that this is something I should use in my own experiments.


Question: What is the computational overhead of this method, compared to the standard approach?

---

> ### Author Response · Authors · 2025-11-25
> **Response to Reviewer Ag39**
>
> We sincerely thank the reviewer for their thoughtful feedback and insightful questions. We address their concerns both in the common response section and in the specific responses below.
>
> *1. I will increase my rating to at least 6 if proper confidence intervals are included.*
>
> We added Confidence Intervals for Tables 1, 2, 3 and 4 in the paper, which evaluate the discriminative performance over test dataset, discriminative performance over Reward Bench, and generative performance as evaluated by UltraRM winrate and AlpacaEval 2.0 for all the alignment algorithms, using Llama3.2-3b model. These confidence intervals are over 6 seeds (including the results from the earlier submitted version of the paper).
>
> *2. Beyond that, I would appreciate seeing a more thorough experimental validation, as its current state does not convince me that this is something I should use in my own experiments.*
>
> We have additionally added experimental results where the PoP annotation was gathered from an LLM evaluator (GPT-4.1-mini), instead of synthetic evaluation, in Section 4.2. We generate the PoP annotation by a random sampling strategy and evaluated the resulting DPO-PoP-Random approach against the other baselines. We show similar results as the synthetic evaluation setting, wherein DPO-PoP-Random significantly outperforms all other baselines with respect to generative performance
>
> *3. What is the computational overhead of this method, compared to the standard approach?*
>
> The DPO-PoP approaches require one additional forward pass, compared to DPO, to infer the margin information from the weaker preferences in the preference over preference data. Another source of computational overhead would be the data scale factor $k$ i.e the amount of preference over preference data the user would like to generate. For example, when $k=2$, we need to iterate over a PoP dataset of size twice the original dataset.

---

### Official Review · Reviewer_p2ui · 2025-10-26

**Soundness:** 2
**Presentation:** 3
**Contribution:** 2
**Rating:** 2
**Confidence:** 3

**Summary:**

The paper proposes a adaptive margin variant of DPO method leveraging “preference-over-preference” (PoP) signals—i.e., which of two preference pairs shows a stronger distinction. The paper explores two different ways of constructing the PoP preference datasets, 1) iterative sampling that represents weak pairs with similar frequencies as original preference datasets; it boosts discriminative accuracy but can hurt generation; 2) randomly sampling where strong preferences naturally appear more; it shows both better overall rewardbench performance and better generation. In summary, the authors show that turning that ordinal signal into adaptive, per-example margins inside DPO (DPO-PoP objective) improves both rewardbench performance and real generation quality.

**Strengths:**

Originality: The paper provides a novel way to get adaptive margins for DPO training by leveraging “preference-over-preference” (PoP) signals.

Quality: The paper provides detailed and easy to understand theory derivation. The paper also provides quite thorough experiments with necessary baselines.

Clarity: The figures and tables are otherwise easy to read except for the ones mentioned in weakness.

Significance: The paper provides a solution to incorporate preference margin information into DPO training leveraging the existing preference datasets - no need to create new prompt/responses trios.

**Weaknesses:**

1. Overall weakness: In the abstract part, the authors indicate that a big part of their motivation is "many existing methods that use adaptive margins assume access to accurate preference scores, which can be difficult for humans to provide reliably." However, the experiments in the whole paper uses Ultrafeedback, a dataset with exact score rating. The authors do claim that they use the ground truth margin as a proxy since using annotator is expensive, but I think it is necessary to do if this is a core advantage that the authors claim to have for their method. Using LLM-as-a-judge to rate the PoP could be a cheaper alternative for example. At least the authors can show what a realistic intended usage case of their method would look like (they intend people to use it when there is no ground truth score rating).
2. Figure 2: The lines are really close but not exactly overlapping. I would like to be able to compare them but the current presentation makes it hard to read. Perhaps the authors can expand the y axes around that range?
3. Section 4.4: Seeing win rate is nice but it's hard to make sense of what the number means without looking at some generation samples.

**Questions:**

1. Regarding weakness 1 mentioned. I think it might be quite infeasible to repeat the experiments with human (or LLM) labeling as opposed to what's used in the paper (actual score margin), but I think it's quite necessary to show if the authors want to claim the strength of the method is no need of accurate preference scores. If the authors can provide a convincing claim to justify the design choice or modify the claim I could consider raising the score.
2. See weakness 2
3. Regarding weakness 3 mentioned: it would be helpful if the authors could provide generation case studies on each representative models.
4. "Interestingly, even though DPO-margin-gt has access to the actual ground-truth margin values, it performs worse in terms of test classification accuracy." I appreciate the comparison with the ground truth margin baseline. This is an interesting observation, can authors provide any intuition on why this could happen?
5. What kind of samples are in Chat Hard? Can authors provide any intuition on why both methods drop considerably on this rewardbench subset?

---

> ### Author Response · Authors · 2025-11-25
> **Response to Reviewer p2ui**
>
> We sincerely thank the reviewer for their thoughtful feedback and insightful questions. We address their concerns both in the common response section and in the specific responses below.
>
> *1. Regarding weakness 1 mentioned. I think it might be quite infeasible to repeat the experiments with human (or LLM) labeling as opposed to what's used in the paper (actual score margin), but I think it's quite necessary to show if the authors want to claim the strength of the method is no need of accurate preference scores. If the authors can provide a convincing claim to justify the design choice or modify the claim I could consider raising the score.*
>
> We gather PoP annotations from an LLM (GPT-4.1-mini), using random sampling, and rerun the suite of experiments performed on the syntetic PoP data created from the Ultrafeedback dataset. These results, in Section 4.2, show that POP approaches significantly improve over other baselines with respect to generative quality.
>
> *2. Figure 2: The lines are really close but not exactly overlapping. I would like to be able to compare them but the current presentation makes it hard to read. Perhaps the authors can expand the y axes around that range?*
>
> We have updated Figure 1a to include an inset zoomed into the range 0.70–0.72, allowing clearer comparison between methods that previously appeared to overlap.
>
> *3. Regarding weakness 3 mentioned: it would be helpful if the authors could provide generation case studies on each representative models.*
>
> We provide generation samples for each of the models, on the test set, in Appendix J. From the results, we observe that DPO-PoP-Random appears to be slightly more coherent and informative than the other approaches.
>
> *4."Interestingly, even though DPO-margin-gt has access to the actual ground-truth margin values, it performs worse in terms of test classification accuracy." I appreciate the comparison with the ground truth margin baseline. This is an interesting observation, can authors provide any intuition on why this could happen?*
>
> We hypothesize that this is on account of POP methods infering continuous fine-grained margins that are more accurate, compared that DPO-margin-gt that uses rating scaling annotations, that have finite resolution and can be noisy, improperly calibrated [1].
>
> [1] Using natural language explanations to rescale human judgments, Wadhwa et al
>
> *5. What kind of samples are in Chat Hard? Can authors provide any intuition on why both methods drop considerably on this rewardbench subset?*
>
> This is description for Chat Hard, taken fromthe RewardBench paper [2]
>
> "Testing a reward model’s abilities to understand trick questions and subtly different instruction responses. Prompts and chosen, rejected pairs are selected from MT Bench examples with similar ratings and adversarial data specifically for fooling LLM-as-a-judge tools from LLMBar’s evaluation set"
>
> From Table 2, we observe that Vanilla-DPO achieves the strongest performance on the Chat Hard subset among all baselines. In contrast, DPO-PoP-Iter performs the worst, followed by DPO-margin-gt and DPO-PoP-Random, which exhibit similar performance levels. Notably, the adaptive-margin baselines, those that place the margin inside the log-sigmoid, show a consistent drop in performance on Chat Hard. We hypothesize that this degradation arises from imposing margins derived from the UltraFeedback dataset, whose preference structure may not align well with the Chat Hard domain. DPO-PoP-Iter performs particularly poorly due to overfitting to weak-preference examples in UltraFeedback, consistent with the explanation and analysis presented in the main paper.
>
> [2] RewardBench: Evaluating Reward Models for Language Modeling, Lambert et al

---

> > ### Comment · Reviewer_p2ui · 2025-11-26
> >
> > I thank the authors for the detailed response and additional experiment. I would still like to follow up on question 1 but fully resolved for all the other questions. I appreciate the additional experiments that demonstrate the method still works for generated labels. I just find the paper writing needs to be a bit restructured. If not needing "accurate preference scores" is the emphasis and a major contribution of this method, this set of new experiment would be more convincing than using the ground truth label, thus could potentially be more appropriate to put in main? Otherwise, you could deemphasize that point from abstract and intro. I am happy to discuss on this point and will increase my score to 4 if this remaining part can get addressed or more detailed explanation.

---

> > > ### Author Response · Authors · 2025-11-27
> > >
> > > We appreciate the reviewer for engaging thoughtfully with our rebuttal and providing valuable feedback.
> > >
> > > We agree that experiments using LLM-annotated preference-over-preference data would strongly support our claim that accurate preference scores are not required. We also concur that these results are better suited for the main text, and have therefore moved this section from the Appendix to Section 4.2 in the paper.

---

### Official Review · Reviewer_yex2 · 2025-10-28

**Soundness:** 2
**Presentation:** 2
**Contribution:** 3
**Rating:** 4
**Confidence:** 5

**Summary:**

This paper proposes a new method to incorporate a margin into the RLHF objective by introducing a novel data collection strategy "Preference-over-Preference". Instead of standard pairwise comparisons, this method asks labelers to compare two pairs of responses to determine which pair exhibits a more significant difference. The authors demonstrate that this approach, when integrated with DPO, is effective compared to vanilla DPO and DPO with other margin-based baselines. The paper also explores two sampling strategies for the PoP dataset, revealing an important trade-off between the model's discriminative and generative performance.

**Strengths:**

- The proposed "Preference-over-Preference" data collection and objective are novel..
- The experimental evaluation is comprehensive, analyzing various aspects of model performance.

**Weaknesses:**

1. Reliability of the Evaluator (UltraRM): The trustworthiness of UltraRM as the primary evaluator is not sufficiently established. UltraRM is trained on GPT-4 annotations (UltraFeedback) and it is not clear how strong this judge is. The paper should include quantitative metrics on the UltraFeedback test set, such as the correlation or agreement rate between UltraRM's predictions and the original GPT-4 annotations. This is crucial for validating UltraRM as a reliable evaluation tool. It might be good to directly use GPT-4 as the judge, since the dataset is annotated by GPT-4.
2. Lack of Optimization Dynamics and KL Analysis: The paper provides no results of model behaviors during the optimization process. Incorporating a margin can sometimes introduce optimization instability, yet no training curves (e.g., win rate or accuracy vs. training steps) are provided. Crucially, the KL divergence from the reference policy during training is not reported. It is difficult to assess the results fairly, as a model's superior accuracy might simply be a consequence of a much higher KL divergence (i.e., straying further from the reference policy). The authors should provide plots (similar to Figure 1 in Tang et al., 2024) showing the evolution of win rate, accuracy, and KL divergence throughout the training process.
3. Baseline Comparisons: The DPO-PoP-random method employs datasets sampled based on the ground-truth margin. This sampling strategy may implicitly provide the PoP models with additional information, as the resampling alters the distribution of margins in the training set. As an ablation, one more baseline DPO should be trained on the same set of preferred pairs derived from the PoP-random dataset (effectively treating 0 as the margin).

[1] Tang et al 24: arXiv:2503.19618

**Questions:**

1. Please clarify the dataset used for calculating the results in Table 1. Is it the UltraFeedback test set?
2. Appendix E shows that different values of $k$ affect the trade-off between discriminative and generative performance. This appears to be a key hyperparameter. Which value of $k$ was used to generate the main results in Tables 1 and 2? This information is critical and should be included in the main paper. Was the same $k$ used for all experiments reported in these tables?
3. Line 438 states that the method leads to "significant gains" for “when the size of the preference dataset is small” Please clarify what evidence you have to justify this claim.
4. Regarding the label noise analysis in Appendix E:
    - The current method simulates noise by randomly flipping labels with a constant probability. A more naturalistic noise model might be to sample the PoP using a Bradley-Terry (BT) model, where a temperature-like constant could control the noise level.
    - In Figure 3, could you add the case for label noise level $p=0$ (i.e., no noise)?

---

> ### Author Response · Authors · 2025-11-26
> **Response to Reviewer yex2 - Part 1**
>
> We sincerely thank the reviewer for their thoughtful feedback and insightful questions. We address their concerns both in the common response section and in the specific responses below.
>
> *1. Reliability of the Evaluator (UltraRM)*
>
> UltraRM is a larger model trained on the Ultrafeedback dataset, which is why we used this as a GOLD RM for evaluation. This is similiar in spirit, to papers on Scaling Laws which used a larger Gold RM for evaluation [1]. Using GPT-4 as the judge would be expensive. This is on account of the test data having 2000 datapoints. So across 6 methods, 2 models and ablations for noise and dataset size, this would make it very expensive. To quell any doubts about the quality of UltraRM as an evaluator, we compute the agreement rates between GPT4 and UltraRM and provide the results along with confidence intervals estimated from bootstrap resampling. We compared the generations for a subset of 512 prompts from the Ultrafedback test set, from DPO-PoP-Random and Vanilla DPO using GPT-4.1-mini as an evaluator. We then used Bootstrap Resampling to estimate the mean and standard deviation of the agreement rate between GPT-4.1-mini and UltraRM as a judge. Over 100,000 resampled datasets, we observed a mean agreement rate of 69.03% and standard deviation of 2.18%. This implies that UltraRM serves as a reasonable proxy for GPT-4.1-mini to evaluate model generations.
>
> [1] Scaling Laws for Reward Model Overoptimization, Gao et al
>
> *2. Lack of Optimization Dynamics and KL Analysis*
>
> We provide training curves that show the evolution of test classification accuracy on the Ultrafeedback test set, KL Divergence wrt reference policy, and UltraRM winrates of model generations in Appendix G.
>
> *3. Baseline Comparisons*
> The DPO-PoP-random method employs datasets sampled based on the ground-truth margin. This sampling strategy may implicitly provide the PoP models with additional information, as the resampling alters the distribution of margins in the training set. As an ablation, one more baseline DPO should be trained on the same set of preferred pairs derived from the PoP-random dataset (effectively treating 0 as the margin).
>
> DPO-PoP-Random randomly samples pairs of preferences and infers PoP annotation from the margin information in the Ultrafeedback dataset (in the synthetic setup). As enquired, we provide an ablation for a variant that performs Vanilla DPO on the PoP dataset (i.e ignoring magin). We refer to this variant as DPO-resampled. We use the already existing results in the paper for all approaches, except DPO-resampled. The following tables show the test classification accuracy and UltraRM winrate for this approach in comparision to other baselines.
>
> | Method             | Test Accuracy | Win Rate (%) |
> |--------------------|------------------|--------------|
> | Vanilla DPO       |      71%          | - |
> | DPO-margin-1       |     71%       | 55%          |
> | DPO-margin-gt      |     71%      | 63%          |
> | DPO-margin-gt-scaled |   72%      | 54%          |
> | DPO-PoP-iter       |     79%      | 57%          |
> | **DPO-PoP-random** |     71%   | **64%**      |
> | *DPO-resampled*|  71%      | 61%   |
>
> We observe that DPO-PoP approaches and DPO-margin-gt still outperform DPO-resampled in both test classification accuracy and UltraRM win rate, highlighting the importance of adaptive margins for improving performance.

---

> ### Author Response · Authors · 2025-11-26
> **Response to Reviewer yex2 - Part 2**
>
> We sincerely thank the reviewer for their thoughtful feedback and insightful questions. We address their concerns both in the common response section and in the specific responses below. We continue from Part 1.
>
> ### Response to Questions
> *1. Please clarify the dataset used for calculating the results in Table 1. Is it the UltraFeedback test set?*
>
> Yes, we use the Ultrafeedback test set to evaluate classification accuracy in Table 1.
>
> *2. Appendix E shows that different values of \\( \alpha \\) affect the trade-off between discriminative and generative performance. This appears to be a key hyperparameter. Which value of \\( \alpha \\) was used to generate the main results in Tables 1 and 2? This information is critical and should be included in the main paper. Was the same \\( \alpha \\) used for all experiments reported in these tables?*
>
> We believe the reviewer is referring to the data scale multipler $k$ here, not $\alpha$. We used a value of $k=2$ for all experiments, expect for ablations, wherever we mention the value of $k$ used. So $k=2$ was used in Table-1 and Table-2. We mentioned this in the hyperparameters for our experiments in Appendix B. We agree that this is important information and have added it to the main section of the paper.
>
> *3. Line 438 states that the method leads to "significant gains" for “when the size of the preference dataset is small.” Please clarify what evidence you have to justify this claim.*
>
> We provide a study of the impact of data scale $k$ in Appendix E.2. In order to perform this ablation, we used a smaller subset of the Ultrafeedback train data, so that we can study the effect of different values of $k$, which would have been infeasible when working with the entire Ultrafeedback data (that comprises 60k datapoints). The cited statement is refering to Table 15 in the paper, where DPO-PoP-Random achieved a win rate of 77% over Vanilla DPO at $k=4$, whereas DPO-margin-gt that uses noisy margin information from rating scale annotations achieves a winrate of only 60%.
>
> *4. Regarding the label noise analysis in Appendix E:
>    - The current method simulates noise by randomly flipping labels with a constant probability. A more naturalistic noise model might be to sample the PoP using a Bradley-Terry (BT) model, where a temperature-like constant could control the noise level.
>    - In Figure 3, could you add the case for label noise level 0 (i.e., no noise)?*
>
> We have completed noise experiments using BT-model labels with a temperature parameter controlling the noise level. The results are reported below.
>
> The table below reports test classification accuracy for the two PoP methods across different temperature values. We observe that accuracy generally increases with temperature. Under temperature-based sampling, weaker preference-over-preference signals are flipped more often than stronger ones, which we hypothesize introduces a regularizing effect that contributes to the performance gains.
>
> | Method         | Temp-1 | Temp-2 | Temp-4 | Temp-8 | Temp-16 |
> |---------------|--------|--------|--------|--------|---------|
> | DPO-PoP-Iter      | 0.8060 | 0.8222 | 0.8343 | 0.8327 | 0.8428  |
> | DPO-PoP-Random    | 0.6720 | 0.6841 | 0.6927 | 0.6826 | 0.6952  |
>
> The table below reports UltraRM win rates for the two PoP methods across different temperature values. For DPO-PoP-Iter, generative performance remains relatively stable as temperature increases. In contrast, DPO-PoP-Random shows a slight upward trend in performance with higher temperatures.
>
> | Method            | Temp-1 | Temp-2 | Temp-4 | Temp-8 | Temp-16 |
> |-------------------|--------|--------|--------|--------|---------|
> | DPO-PoP-Iter      | 0.56   | 0.54   | 0.56   | 0.55   | 0.56    |
> | DPO-PoP-Random    | 0.56   | 0.56   | 0.56   | 0.57   | 0.59    |
>
>
> We have added noise=0 results to Figures 3 and 4. This only involved adding in results from the main section of the paper, where there was no noise.

---

### Official Review · Reviewer_NwPi · 2025-10-29

**Soundness:** 3
**Presentation:** 3
**Contribution:** 3
**Rating:** 4
**Confidence:** 3

**Summary:**

The paper proposes DPO-PoP, an extension of Direct Preference Optimization (DPO) that introduces adaptive margins based on preferences over preferences (PoP): annotations indicating which preference pair expresses a stronger preference contrast. Instead of relying on scalar human or model scores, DPO-PoP infers relative margin strengths through ordinal supervision. The approach integrates this information into the DPO loss using stability enhancements such as margin clipping and Polyak-averaged targets. Experiments on UltraFeedback, RewardBench, and AlpacaEval 2.0 show that DPO-PoP variants outperform standard and fixed-margin DPO baselines. The paper also reports an interesting tradeoff: improving discriminative (classification) performance may reduce generative quality, and vice versa.

**Strengths:**

1.	Simple yet coherent extension. The proposed method integrates adaptive margins into DPO cleanly, requiring no extra reward model or RL loop.
2.	Comprehensive empirical evaluation. The experiments are extensive, covering multiple model scales, datasets, and sensitivity factors (sampling, scale, label noise).
3.	Practical annotation framing. The use of ordinal comparisons rather than numeric ratings is a plausible direction for more robust human feedback.
4.	Clear empirical takeaway. The study highlights a consistent tradeoff between discriminative and generative alignment, which is valuable for practical RLHF tuning.
5.	Readable and reproducible. Code availability, training details, and hyperparameters are well documented.

**Weaknesses:**

1.	Limited empirical gains. The improvements over baselines are modest and might not justify the additional supervision and implementation complexity.
2.	Synthetic validation. All PoP data are simulated from existing scalar scores; thus, the central claim that human PoP labels are easier or more reliable is not empirically validated.
3.	Incremental novelty. The technique mainly reuses known margin-scaling principles, framed as "preference over preference." The conceptual jump beyond prior margin-based DPO methods is small.
4.	Lack of theoretical insight. The paper would be more robust if they can explain why PoP improves discriminative calibration or how the discriminative–generative tradeoff arises.
5.	Scalability of data collection. If PoP labels were collected directly, the quadratic comparison cost could make large-scale annotation expensive unless approximate or active-sampling methods are introduced.

**Questions:**

1.	Have you tested DPO-PoP with actual human PoP annotations rather than synthetic scores?
2.	Can you formalize or visualize the observed discriminative–generative tradeoff? What governs the balance between the two?
3.	How scalable is PoP labeling in real-world RLHF datasets with millions of preferences? Could adaptive sampling mitigate the quadratic cost?
4.	The observed improvements are relatively small—what concrete aspects of model behavior improve qualitatively under DPO-PoP?

---

> ### Author Response · Authors · 2025-11-25
> **Response to Reviewer NwPi - Part 1**
>
> We sincerely thank the reviewer for their thoughtful feedback and insightful questions. We address their concerns both in the common response section and in the specific responses below.
>
> **Novelty.** We first clarify that Preference-over-Preference (PoP) does *not* introduce implementation complexity beyond standard DPO. The training pipeline is identical to DPO, with only a single additional forward pass used to infer continuous margins from weaker preferences. Importantly, in PoP we simply elicit pairwise comparisons rather than a rating-scale score, used by conventional margin based methods. PoP infers preference strength directly from comparisons rather than relying on coarse, discrete human-provided ratings.
>
> Conceptually, the novelty is that PoP replaces rating-scale–based strength annotations, which are known to be noisy and inconsistent, with *pairwise* strength inference. To the best of our knowledge, prior margin-based DPO work has always relied on fixed margins or margins derived from rating scales. No prior work proposes using pairwise comparisons themselves to infer continuous margins, nor analyzes the implications of these inferred margins for generalization. We also added a discriminative–generative tradeoff analysis in Appendix I and new generalization bounds for adaptive-margin classifiers in Appendix H; this theoretical perspective does not appear in prior DPO margin-scaling work.
>
> **Empirical gains.** We also emphasize that PoP achieves more than modest improvements in several settings:
>
> - **Low-data regime:** As shown in Appendix E.2, PoP methods significantly outperform baselines in generative quality when preference data is limited, a highly practical regime where existing methods degrade sharply.
> - **Generalization and robustness:** Tables 2,3,4,7,8 show that DPO-PoP-Random consistently improves over all baselines on out-of-domain evaluations (RewardBench, Model Generations, AlpacaEval 2.0), highlighting improved robustness and cross-domain reliability.
> - **Improved In-domain discriminative ability:** Table 1 shows that DPO-PoP-Iter yields substantially higher test classification accuracy, indicating improved alignment with the underlying human preference structure.
> - **Annotation practicality:** Section 4.2 shows that PoP remains effective even when PoP annotations are produced by an LLM annotator. This has real world implications, as we show that POP methods, DPO-PoP-Random in particular, improves generative performance in comparision to conventional baselines that use fixed/no margins or noisy adaptive margins obtained from rating scale annotations.
> - **Noise robustness:** Appendix F demonstrates that PoP maintains stable performance under annotation noise.
>
> Taken together, these contributions go beyond “incremental margin scaling”: PoP introduces a new and practical way to elicit preference-strength information from pairwise comparisons, improves generalization behavior, and yields significant gains in low-data settings, while being robust to noise. We believe these points constitute meaningful novelty.

---

> ### Author Response · Authors · 2025-11-25
> **Response to Reviewer NwPi - Part 2**
>
> We sincerely thank the reviewer for their thoughtful feedback and insightful questions. We address their concerns both in the common response section and in the specific responses below. We continue from Part-1.
>
> *1. Have you tested DPO-PoP with actual human PoP annotations rather than synthetic scores?*
>
> We gather PoP annotations from an LLM (GPT-4.1-mini), using random sampling, and rerun the suite of experiments performed on the syntetic PoP data created from the Ultrafeedback dataset. These results, in Section 4.2, show that POP approaches significantly improve over other baselines with respect to generative quality.
>
> *2. Can you formalize or visualize the observed discriminative–generative tradeoff? What governs the balance between the two?*
>
> We added this discussion in Appendix I of the paper and provided supporting theory in Appendix H, drawing from classical SVM analysis. The theory characterizes the generalization behavior of adaptive-margin classifiers and shows that the upper bound on generalization error depends on $\widetilde{M} = \sqrt{\sum_{i=1}^{N} \frac{1}{m_i^{2}}}$
>
> This implies that overfitting to datapoints with weak margins (small $m_i$) can significantly worsen the bound and thus hurt out-of-domain generalization. However, completely ignoring weaker preferences can degrade in-domain performance. This is reflected in Table 1, where DPO-PoP-Iter achieves higher test classification accuracy than baselines on data drawn from the same distribution as the training set.
>
> The bound is particularly indicative when the train and test distributions differ more substantially, as in evaluations on RewardBench, Aligned model Generations, and AlpacaEval 2.0. In these settings (Tables 2,3,4,7,8), DPO-PoP-Random outperforms DPO-PoP-Iter and other baselines, which aligns with its tighter generalization bound arising from not incorporating the large fraction of weak-preference (small-margin) datapoints present in DPO-PoP-Iter.
>
>
> *3. How scalable is PoP labeling in real-world RLHF datasets with millions of preferences?** Could adaptive sampling mitigate the quadratic cost?*
>
> We would like to point out Appendix E, that demonstrates that it suffices to use only a small scalar multiple of the dataset size (k) and provided experimental results that study the effect of size of data on performance. Hence the scaling is still linear.
>
> *4. The observed improvements are relatively small — what concrete aspects of model behavior improve qualitatively under DPO-PoP?*
>
> In this paper, we show that substituting rating scale annotations with pairwise comparision annotations, which allows us to infer fine-grained continuous margin information instead of finite discrete values, matches or exceeds the performance of alignment methods that use no/fixed margins or adaptive margins derived from noisy rating scale annotations. We also highlight the improvements obtained by POP methods over conventional baselines below.
>
> - **Improved Generative quality in low-data regimes:** We show in Appendix E.2, that in settings with small amounts of preference data, PoP methods significantly outperform other baseline methods with respect to generative quality.
> - **Better Generalization Ability and Robustness:** Evaluative results in Tables 2,3,4,7,8 show that PoP-Random has better generalization performance and robustness compared to all other baselines.
> - **Better In-Domain Discriminative Ability:** Additionally in Table 1, we show that DPO-PoP-Iter improves in-domain test classification accuracy significantly over all other baselines.
> - **Robustness to Noise:** POP methodes are also fairly robust to labeling noise in POP annotations, as evidenced in Appendix F.
> - **No additional implementation complexity:** Preference over Preference annotation does not have additional implementation complexity over DPO. We just use the same DPO algorithm, but one additional forward pass to infer the margin information from the weaker preference.. Furthermore, similar to how rating based scales (Eg: Likert) are used to gather information about the strength of the preference, we use pairwise preference comparisions using Preference over Preference annotation to infer the strength of preferences. The only distinction is that conventional approaches use Rating scale based annotations whereas we annotate using pairwise comparisions, to infer preference strength.

---

### Comment · Area_Chair_yLwd · 2025-11-25

Dear Reviewers,
Thank you to those who have already begun interacting with the authors — your timely follow-ups are greatly appreciated and reflect the professionalism and care that uphold our community’s standards.
For reviewers who have not yet responded, I would like to offer a gentle reminder. Authors have invested substantial time and effort into preparing their rebuttals, carefully addressing each concern raised in the reviews. As fellow researchers, we all understand the importance of being heard and having our clarifications considered. Even a brief acknowledgment or follow-up question helps ensure that the evaluation remains fair, thorough, and respectful of everyone’s work.
Your engagement during this phase is essential for maintaining a constructive and high-quality review process. Thank you again for your service and for treating both your fellow reviewers and the authors with the same consideration you would hope to receive.
Best regards,
AC

---

### Author Response · Authors · 2025-11-27
**Common Response**

We thank all the reviewers for their detailed and thoughtful feedback on our submission and appreciate the opportunity to clarify their questions. Here we address the common question raised by the reviewers, regarding the use of synthetic data. We also highlight the novelty and contributions of our work, alongside theoretical contributions.

---

### Synthetic Data Evaluation

We collect preference-over-preference (PoP) annotations using an LLM annotator (GPT-4.1-mini) with random sampling. We then evaluate DPO-PoP-Random against standard baselines on both discriminative and generative metrics, in section 4.2. As in our synthetic-data experiments, DPO-PoP-Random exhibits consistent and substantial gains in generative ability, as measured by UltraRM and AlpacaEval 2.0 win rates. These results confirm that our findings are not an artifact of synthetic data but extend to LLM-generated annotations as well.

---

### Novelty & Contributions

We propose Preference over Preference (PoP) annotation to infer adaptive margin information compared to conventional approaches that infer preference strength information from rating scale annotations (Eg: Likert scale). PoP supervision allows us to infer fine-grained continuous margin information, compared to rating scale annotations that are coarse, and can be noisy or uncalibrated [1]. We propose alignment methods that leverage PoP supervision and demonstrated that they lead to improved generative and discriminative performance over conventional approaches that use no/fixed margin information or adaptive margin information derived from noisy rating scale annotations.

**Key empirical findings:**

- **Improved generalization and robustness.**
  Tables 2,3,4,7,8 show that DPO-PoP-Random outperforms all baselines on out-of-domain evaluations (RewardBench, Model Generations, AlpacaEval 2.0), demonstrating stronger robustness and cross-domain reliability.

- **Improved in-domain discriminative accuracy.**
  Table 1 shows that DPO-PoP-Iter achieves substantially higher test classification accuracy.

- **Low-data robustness.**
  In Appendix E.2, PoP methods significantly outperform baselines in generative quality when preference data is limited—a highly practical regime where existing methods degrade sharply.

- **Annotation practicality.**
  Section 4.2 shows that PoP remains effective even when PoP annotations are produced by an LLM annotator. DPO-PoP-Random improves generative quality compared to baselines using fixed/no margins or noisy rating-scale–derived margins, demonstrating real-world feasibility.

- **Noise robustness.**
  Appendix F shows that PoP maintains stable performance under annotation noise, further supporting its practical utility.

[1] *Using natural language explanations to rescale human judgments*, Wadhwa et al.

---

### Discriminative–Generative Tradeoff

We provide a generalization bound for adaptive-margin classifiers in Appendix H, extending classical SVM analysis. The generalization error upper bound depends on  $\widetilde{M} = \sqrt{\sum_{i=1}^{N} \frac{1}{m_i^{2}}}$,
where $m_i$ are the margins of training data points. This framework explains the observed difference between DPO-PoP-Random and DPO-PoP-Iter.

DPO-PoP-Random achieves a **smaller** $\widetilde{M}$, since it does not incorporate the large fraction of weak-preference (small-margin) comparisons included in DPO-PoP-Iter. As a result, its generalization bound is tighter, particularly when train–test distributions differ substantially.

Empirically, this is reflected in Tables 2,3,4,7,8: DPO-PoP-Random consistently outperforms DPO-PoP-Iter and other baselines on RewardBench, Model Generations, and AlpacaEval 2.0. However, completely ignoring weaker preferences can degrade in-domain performance. This is reflected in Table 1, where DPO-PoP-Iter achieves higher test classification accuracy than baselines, on data drawn from a similar distribution as the training set. This alignment between theory and experiment clarifies the discriminative–generative tradeoff and highlights the role of margin structure in generalization.

---

### Author Response · Authors · 2025-12-03
**Response to Area Chairs-Part 1**

We sincerely thank you and all reviewers for the significant time and effort dedicated to reviewing our work. To facilitate the meta-review process, we briefly summarize our paper's contributions, reviewer discussions, and the paper revisions made during rebuttal.

## Contributions

We propose Adaptive margin RLHF using Preference over Preference (PoP) annotations to infer adaptive margin information compared to conventional approaches that infer preference strength information from rating scale annotations (Eg: Likert scale). PoP supervision allows us to infer fine-grained continuous margin information, compared to rating scale annotations that are coarse, and can be noisy or uncalibrated. We propose alignment methods (DPO-PoP) that leverage PoP supervision and demonstrated that they lead to improved generative and discriminative performance over conventional approaches that use no/fixed margin information or adaptive margin information derived from noisy rating scale annotations.

We showed DPO-PoP methods provide:

- **Improved generalization and robustness.**
    Tables 2,3,4,7,8 show that DPO-PoP-Random outperforms all baselines on out-of-domain evaluations (RewardBench, Model Generations, AlpacaEval 2.0), demonstrating stronger robustness and cross-domain reliability.

- **Improved in-domain discriminative accuracy.**
    Table 1 shows that DPO-PoP-Iter achieves substantially higher test classification accuracy.

- **Low-data robustness.**
    In Appendix E.2, PoP methods significantly outperform baselines in generative quality when preference data is limited—a highly practical regime where existing methods underperform.

- **Annotation practicality.**
    Section 4.2 shows that PoP remains effective when PoP annotations are produced by an LLM annotator. DPO-PoP-Random improves generative quality compared to baselines using fixed/no margins or noisy rating-scale–derived margins, demonstrating real-world feasibility.

- **Noise robustness.**
    Appendix F shows that PoP maintains stable performance under annotation noise, further supporting its practical utility.

## Reviewer Discussions

All reviewers appreciated the simplicity of the proposed PoP approaches and the thorough empirical evaluation.

The primary concern across most reviewers was the use of synthetically annotated PoP data in our experiments. We added experimental results that use LLM-annotated PoP data, in Section 4.2, and demonstrated similar results to the synthetic setting, with DPO-PoP-Random improving generative performance over all other baselines.

- **Reviewer NwPi**

    1) We provided experimental results in Section 4.2, using LLM annotated PoP data, to address the pointed weakness about the use of synthetic data
    2) We provided theoretical justification for the observed discriminative-generative tradeoff, in Appendix H and I, to address their comment on lack of theoretical insights.
    3) We corrected the reviewers misunderstanding on quadratic scalability costs, and also reiterated the novelty and contributions of our paper.

- **Reviewer yex2**

    1) To address the reviewer's comment on the use of UltraRM as an evaluator, we provide justification and also agreement rates between UltraRM and GPT4. UltraRM and GPT-4.1 have an agreement rate of ~70% justifying the usage of UltraRM as a reasonable proxy for GPT.
    2) To address the comment on lack of optimization dynamics, we provided the evolution of test classification accuracy, KL divergence and winrates over training in Appendix G.
    3) We also provided results for an additional baseline, as well as noise experiments that vary the temperature of the BT preference model, in our response to the reviewer.

- **Reviewer p2ui**

    1) We provided experimental results in Section 4.2, using LLM annotated PoP data, to address the pointed weakness about the use of synthetic data
    2) We also provided Generation case-studies in Appendix J, to address their comment on not providing generative samples for the approaches studied in the paper.

    **We would like to point that Reviewer p2ui, were about to improve their score**, before the unfortunate incident on Nov 27 happened.

- **Reviewer Ag39**

    1) **The reviewer mentioned that they would increase their score to at least a 6 if confidence intervals were provided**. We updated Tables 1,2,3, and 4 to include confidence intervals over 6 seeds, as requested.
    2) We provided experimental results in Section 4.2, using LLM annotated PoP data, to address the pointed weakness about the use of synthetic evaluation.

We hope the Area Chairs will take into account that we provided substantial results and explanations addressing the weaknesses and questions raised by the reviewers, especially given that we did not receive responses from most of the reviewers.

---

> ### Author Response · Authors · 2025-12-03
> **Response to Area Chairs-Part 2**
>
> We continue from Part-1.
>
> ## Revision Summary
> - Confidence Intervals added to Tables 1,2,3, and 4
> - Experiments using LLM annotated PoP data in Section 4.2
> - Evolution of metrics in Appendix G
> - Theory discussing the generalization performance of Adaptive-Margin classifiers in Appendix H
> - Discussion on the Discriminative-Generative tradeoff in Appendix I
> - Generation Case-studies in Appendix J
> - Prompt used to query the LLM for PoP Annotation in Appendix K

---

### Meta-Review · Area_Chair_TAkK · 2026-01-06

**Summary:**

This paper studies adaptive margin RLHF using Preference over Preference (PoP) annotations, which allows us to infer fine-grained continuous margin information. PoP was then integrated with DPO as an alignment approach with improved generalization and robustness performance. As an empirical paper, the simplicity of the approach was appreciated. However, there were significant concerns regarding the use of the synthetic PoP data, as well as the thoroughness of the experiments. The rebuttal and new experiments have helped address some of the concerns. However, clearing the bar for ICLR requires significant editing of the paper. I recommend that the authors incorporate the feedback from this round in preparing the next version of the paper.

**Reviewer Concerns:**

There were some concerns regarding the use of the synthetic data, the novelty of the technical contributions, and the lack of theoretical insights/guarantees, which have been partially, though not fully, addressed by the rebuttal and the new experiments.

**Reviewer Scores:**

It is likely that Reviewers p2ui and Ag39 will increase the score after the discussion, and in fact, Reviewer p2ui was about to improve their score.

---

### Decision · Program_Chairs · 2026-01-26

Reject